# EARLINET observations of Saharan dust intrusions over the northern Mediterranean region (2014-2017): Properties and impact in radiative forcing

Ourania Soupiona[1], Alexandros Papayannis[1], Panagiotis Kokkalis[2], Romanos Foskinis[1], Guadalupe Sánchez Hernández[3,4], Pablo Ortiz-Amezcua[3,4], Maria Mylonaki[1], Christina-Anna Papanikolaou[1], Nikolaos Papagiannopoulos[5], Stefanos Samaras[6], Silke Groß[7], Rodanthi-Elisavet Mamouri[8,9], Lucas Alados-Arboledas[3,4], Aldo Amodeo[5], Basil Psiloglou[10]

[1]School of Applied Mathematics and Physical Sciences, Dept. of Physics, National Technical University of Athens, 15780, Greece
[2]Department of Physics, Kuwait University, Safat, 13060, Kuwait.
[3]Department of Applied Physics, University of Granada, Granada,18071, Spain
[4]Andalusian Institute for Earth System Research, Granada, 18006, Spain
[5]Consiglio Nazionale delle Ricerche, Istituto di Metodologie per l'Analisi Ambientale, Tito Scalo, 85050, Italy
[6]German Aerospace Center (DLR), German Remote Sensing Datacenter (DFD), Wessling, Germany
[7]Institute of Atmospheric Physics, Deutsches Zentrum für Luft- und Raumfahrt (DLR), Oberpfaffenhofen, 82234, Germany
[8]Cyprus University of Technology, Dep. of Civil Engineering and Geomatics, Limassol, Cyprus
[9]ERATOSTHENES Centre of Excellence, Limassol, Cyprus
[10]Institute for Environmental Research and Sustainable Development, National Observatory of Athens, Palaia Penteli, Athens, Greece

* Correspondence to: Ourania Soupiona (raniaphd@mail.ntua.gr)

## Abstract

Remote sensing measurements of aerosols using depolarization Raman lidar systems from 4 EARLINET (European Aerosol Research Lidar Network) stations are used for a comprehensive analysis of Saharan dust events over the Mediterranean basin in the period 2014—2017. In this period, 51 dust events regarding the geometrical, optical and microphysical properties of dust were selected, classifying them and assessing their radiative forcing effect on the atmosphere. From West to East, the stations of Granada, Potenza, Athens and Limassol were selected as representative Mediterranean cities regularly affected by Saharan dust intrusions. Emphasis was given on lidar measurements in the visible (532 nm) and specifically on the consistency of the particle linear depolarization ratio ($\delta_{p532}$), the extinction-to-backscatter lidar ratio ($LR_{532}$) and the Aerosol Optical Thickness ($AOT_{532}$) within the observed dust layers. We found mean $\delta_{p532}$ values of 0.24±0.05, 0.26±0.06, 0.28±0.05 and 0.28±0.04, mean $LR_{532}$ values of 52±8 sr, 51±9 sr, 52±9 sr and 49±6 sr, and mean $AOT_{532}$ values of 0.40±0.31, 0.11±0.07, 0.12±0.10 and 0.32±0.17, for Granada, Potenza, Athens and Limassol, respectively. The mean layer thickness values were found to range from ~1700 to ~3400 m a.s.l. Additionally, based also on a previous aerosol type classification scheme provided by airborne High Spectral Resolution Lidar (HSRL) observations and on air mass backward trajectory analysis, a clustering analysis was performed in order to identify the mixing state of the dusty layers over the studied area. Furthermore, a synergy of lidar measurements and modeling was used to deeply analyze the solar and thermal radiative forcing of airborne dust. In total, a cooling behavior in the solar range and a significantly lower heating behavior in the thermal range was estimated. Depending on the dust optical and geometrical properties, the load intensity and the solar zenith angle (SZA), the estimated solar radiative forcing values range from -59 to -22 W m$^{-2}$ at the surface and from -24 to -1 W m$^{-2}$ at the top of the atmosphere (TOA). Similarly, in the thermal spectral range these values range from +2 to +4 W m$^{-2}$ for the surface and from +1 to +3 W m$^{-2}$ for the TOA. Finally, the radiative forcing seems to be inversely proportional to the dust mixing ratio, since higher absolute values are estimated for less mixed dust layers.

## 1. Introduction

The Saharan desert is one of the major dust sources globally, with dust advections to the Mediterranean countries to be modulated by meteorology along rather regular seasonal patterns (Mona et al., 2012). For instance, in the Western Mediterranean region, the African dust occurrence is higher in summer (Salvador et al., 2014), even though some extreme events might also occur in winter (e.g. Cazorla et al., 2017; Fernández et al. 2019) while in the central Mediterranean region, spring and summer are, usually, associated with dust aerosol loads extending up to altitudes of 3—4 km (Barnaba and Gobbi, 2004). In the Eastern Mediterranean, the main dust transport occurs from spring to autumn (Papayannis et al., 2009; Nisantzi et al., 2015; Soupiona et al., 2018) as a result of the high cyclonic activity over the northern Africa during these periods (Flaounas et al., 2015). Considering also that the Mediterranean basin is a region of high evaporation, low precipitation and remarkable solar activity, the transportation of aerosols accompanied with aging and mixing processes make this area a study of interest for present and future climate change effects (Michaelides et al., 2018).

It is well documented that mineral dust highly influences the atmospheric radiative balance through scattering and absorption processes (direct effect) as well as cloud nucleation, formation and lifetime (indirect effects) as summarized in IPCC (2014). Considerable uncertainties in quantifying the global direct radiative effects of aerosols arise from the variability of their spatio-temporal distribution and the aging/mixing processes that can affect their optical, chemical and microphysical properties and influence many processes that modulate regional climate. Therefore, the magnitude and even the sign of the dust aerosol solar radiative forcing are highly uncertain as they strongly depend on their optical properties, their size distribution and their complex refractive index (CRI) values. Papadimas et al. (2012) reported that the aerosol optical depth seems to be the main parameter for modifying the regional aerosol radiative effects (under cloud-free conditions) and, that on an annual basis, aerosols can induce a significant "planetary" cooling over the broader Mediterranean basin. Other studies (Quijano et al., 2000; Tegen et al., 2010) have shown that the presence of clouds and the surface albedo are also unquestionable parameters affecting the net solar radiative transfer at the top of the atmosphere. However, a comprehensive analysis from ground-based aerosol optical properties to vertical profiles of short- and long-wave radiation estimations in the Mediterranean region has been reported so far only in a few papers (Sicard et al., 2014; Meloni et al., 2003; 2015; Valenzuela et al., 2017; Gkikas et al. 2018).

Although there have been a lot of studies about Saharan dust optical properties based on the lidar technique (Landulfo et al., 2003; Ansmann et al., 2009; Papayannis et al., 2009; Córdoba-Jabonero et al., 2011; Tesche et al., 2011; Mona et al., 2012; Groß et al., 2013; Navas-Guzmán et al., 2013; Granados-Muñoz et al., 2016; Mandija et al., 2016, 2017; Rittmeister et al., 2017; Soupiona et al., 2018), systematic long-term statistical studies are quite scarce since few aerosol depolarization data are available covering long periods. Saidou Chaibou et al., (2020) address the importance of dust effects in climate studies in order to improve the accuracy of climate predictions. As they mention, even if improved assessment of dust impact on climate requires continuous observations from both satellites, and ground-based instrument networks, the use of climate models is also crucial to improve our understanding of dust distribution, its properties and its impact in radiation budget. In an earlier study, Pérez et al. (2006) proposed that a regional atmospheric dust model, with integrated dust and atmospheric radiation modules, represents a promising approach for further improvements in numerical weather prediction practice and radiative impact assessment over dust-affected areas, especially in the Mediterranean. Hence, an in-depth study of the role of aerosols on the radiative forcing into different regions in the Mediterranean basin is still needed. While a synergy of ground-based lidar measurements and modelling seems very promising for obtaining radiative forcing estimations of dust aerosols, the use of inputs from regional models could also contribute for such estimations in areas where measurements are unavailable.

This paper aims to fill some of the aforementioned gaps by combining statistical lidar data of aerosol optical and microphysical properties with radiative transfer estimations and is organized as follows. A brief summary of the

four selected EARLINET Mediterranean stations is given in Sect. 2 along with the data selection of the dust cases. Section 3 includes the description of the methodologies applied and the tools and models used for retrieving the aerosol optical and microphysical properties and their radiative forcing. The evaluation of the retrieved aerosol mass concentration profiles and of the ground level radiation are also presented. The results of the aerosol optical, geometrical and microphysical properties of the individual dust layers and the clusters, as well as the relevant radiative forcing calculations over the studied areas are discussed in Sect. 4. Finally, concluding remarks are given in Sect. 5.

## 2. Instrumentation and data

The European Aerosol Research Lidar Network (EARLINET, https://www.earlinet.org/, Pappalardo et al., 2014) established in 2000, provides an excellent opportunity to offer a large collection of quality assured ground-based data of the vertical distribution of the aerosol optical properties over Europe. These measurements meet absolute accuracy standards (Pappalardo et al., 2014) to achieve the desired confidence in aerosol radiative forcing calculations. Currently, the network includes 31 active lidar stations distributed over Europe, providing information of aerosol vertical distributions on a continental scale. In this paper, level 2 data of four stations from the EARLINET database (https://data.earlinet.org/) including aerosol backscatter ($\beta_{aer}$), extinction ($\alpha_{aer}$) coefficient and depolarization ratio ($\delta_{aer}$) profiles as a function of height above mean sea level (a.s.l.) were collected and further analyzed, as described below, to estimate their role in radiative transfer calculations in the Mediterranean region.

### 2.1. EARLINET stations

Four EARLINET stations affected by typical Saharan dust intrusions in the Mediterranean were selected (listed from West to East): Granada (Spain), Potenza (Italy), Athens (Greece) and Limassol (Cyprus). A four year (2014–2017) common period of aerosol depolarization Raman lidar data obtained at 532 nm was selected for this analysis. Table 1 summarizes the basic information about these lidar systems for each location. Except the Limassol station that provides data only at 532 nm, the other three stations are equipped with a multiwavelength lidar system able to provide extensive aerosol properties at multiple wavelengths. Namely three $\beta_{aer}$ (355, 532, 1064 nm) and two $\alpha_{aer}$ (355, 532 nm) as well as aerosol intensive properties namely the backscatter and extinction-related Ångström exponents ($AE_{\alpha355/532}$, $AE_{\beta355/532}$, $AE_{\beta532/1064}$ nm), the lidar ratio (LR), and additionally the linear volume ($\delta_{v532}$) and particle depolarization ratio ($\delta_{p532}$) at 532 nm. By using the Raman technique, as proposed by Ansmann et al., (1992), the $\beta_{aer}$ and $\alpha_{aer}$ vertical profiles can be retrieved, with uncertainties of ∼5–15% and ∼10–25%, respectively (Ansmann et al., 1992; Mattis et al., 2002). Therefore, the corresponding uncertainty of the retrieved lidar ratio values is of the order of 11–30%, while the uncertainty for $AE_\beta$ and $AE_\alpha$ ranges between 0.02-0.04 and 0.03-0.08, respectively, as estimated by propagation error calculations.

Table 1: Station name, location, lidar setup and relevant references of the four selected EARLINET stations.

| Station | Abbreviation | Location | Lidar setup | References |
|---|---|---|---|---|
| Andalusian Institute for Earth System Research, University of Granada, Spain | IISTA-CEAMA, **GRA** | 37.16º N, 3.61º W, elev. 680 m | MULHACEN $3\beta + 2\alpha + \delta_{p532}$ Overlap: 500 m a.g.l. | Guerrero-Rascado et al., 2008; 2009 |
| Consiglio Nazionale delleRicerche – Istituto di Metodologie per l'AnalisiAmbientale, Potenza, Italy | CNR-IMAA, **POT** | 40.60º N, 15.72º E, elev. 760 m | MUSA $3\beta + 2\alpha + \delta_{p532}$ Overlap: 405 m a.g.l. | Madonna et al., 2011 |
| Laser Remote Sensing Unit, National Technical University of Athens, Athens, Greece | LRSU-NTUA, **ATZ** | 37.96º N, 23.78º E, elev. 212 m | EOLE/AIAS $3\beta + 2\alpha + \delta_{p532}$ Overlap: 800 m a.g.l. | Papayannis et al., 2020 |
| Cyprus University of Technology, Limassol, Cyprus | CUT, **LIM** | 34.67º N, 33.04º E, elev. 10 m | Polarisation Raman lidar, $1\beta + 1\alpha + \delta_p$ (532 nm) Overlap: 250 m a.g.l. | Nisantzi et al., 2015 |

## 2.2.    Selection of dust events

Dusty cases analyzed in this study were selected based on the values of the aerosol optical properties $\delta_{p532}$ and $LR_{532}$ measured by lidar (Groß et al., 2013). Since pure dust layers are rare over the Mediterranean cities due to continental contamination by urban, pollution or even biomass burning (BB) aerosols, a sufficiently lower $\delta_{p532}$ value with respect to the pure dust values (e.g. Freudenthaler et al., 2009) should be considered to characterize an aerosol layer as a dusty one. Based on previous studies, the respective $LR_{532}$ values for long-range transported mixtures over the Mediterranean area are expected to range between 35-75 sr (Mona et al., 2006; Papayannis et al., 2008; Tesche et al., 2009; Groß et al., 2011; Ansmann et al., 2012; Nisantzi et al., 2015; Soupiona et al., 2018). Consequently, from the total set of Saharan dust events per station listed in the EARLINET database for the period 2014—2017, we considered for further analysis only the data meeting three basic criteria: a) $\delta_{p532} \geq 0.16$ in the free troposphere, b) 35 sr $\leq$ LR $_{532} \leq$ 75 sr in the free troposphere and c) the thickness of the detected layer to be 500 m, at least. The critical height (in meters a.s.l.) in which the first criterion was met, was considered to be the base of the dust layer. This assumption was deemed necessary to be made since usually, the lofted dust layers cannot be distinguished from the top of the Planetary Boundary layer (PBL), while the presence of urban haze and pollution decreases drastically the $\delta_p$ values down to 0.03—0.10 (Gobbi et al., 2000; Groß et al., 2013). The top of the dust layer was estimated as the height where the signals were similar to the molecular scattering (both $\delta_{p532}$ and $\beta_{532}$ tending to zero) in the free troposphere. For some cases of the Athens station, where depolarization measurements were unavailable, the values of the base and top were calculated from the Raman lidar signals, following the procedure proposed by Mona et al. (2006).

Moreover, a careful investigation of the air mass origin and dust transport path was performed by means of backward trajectory analysis. This analysis was carried out using the HYbrid Single-Particle Lagrangian Integrated Trajectory (HYSPLIT) model (https://ready.arl.noaa.gov/HYSPLIT_traj.php, Stein et al., 2015) together with the GDAS (Global Data Analysis System) meteorological files (spatial resolution of 1° × 1°, every 3 hours) as data input. The kinematic back-trajectories were calculated using the vertical velocity component given by the meteorological model with a 96—120 hours pathway (4—5 days back). Modis/Terra information (https://firms.modaps.eosdis.nasa.gov/map) was also taken into account for the corresponding hot spots of possible fires and thermal anomalies along the trajectories (https://firms.modaps.eosdis.nasa.gov/map, not shown here).

Thus, we ended up with 51 individual cases in total, of half- to one-hour averaged lidar profiles each (15 for Granada, 18 for Potenza, 12 for Athens and 6 for Limassol). For the region of Cyprus, the situation is more complex since Middle East dust outbreaks occur also frequently in addition to the Saharan dust events (Nisantzi et al., 2015; Kokkalis et al., 2018; Solomos et al., 2019). On top of that, dust particles originating from Middle East proved to have different lidar ratio values than the corresponding observations over Saharan desert (Mamouri et al., 2013; Kim et al., 2020). Taking this into account, dust cases over the Limassol station originating from Middle East regions were excluded from our study.

The air mass trajectory analysis based on HYSPLIT for each station reveals the origin of each observed layer (Fig. 1). In the majority of cases, air masses originate from west and northwest Africa (Morocco, Mauritania, Algeria and Tunisia). At a first glance, two occurrences seem to dominate: i) trajectories that travel directly from the source to the observation stations and ii) trajectories that circulate over the Mediterranean or the Atlantic Ocean (for the Granada and Potenza cases), Europe and Balkans or even Turkey (for the Limassol and Athens cases) before reaching the observation stations.

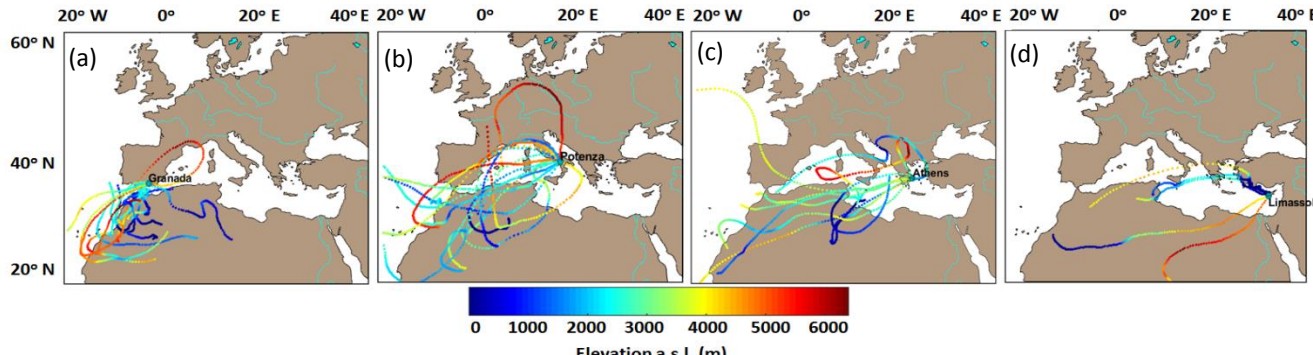

Figure 1: 96—120-hour backward trajectories for air masses arriving over a) Granada, b) Potenza, c) Athens and d) Limassol, for arrival heights of approximately the center of each observed dust layer (51 cases in the period 2014–2017).

## 3.    Methodologies, tools and data evaluation

In order to perform simulations for further investigating the behavior of the transported dust aerosols and their impacts, we used different methods with a variety of tools and models. In this section, we present our efforts
for retrieving vertical dust mass concentration profiles, aerosol microphysical properties and radiative forcing results. The simulations were also partly validated with ground-based radiation measurements.

### 3.1.    Dust mass concentration lidar retrievals

To retrieve the aerosol dust mass concentration profiles, we used the $\beta_{532}$ and the $\delta_{p532}$ coefficients. Furthermore,  by assuming that we have two aerosol types (dust and non-dust) inside the calculated $\beta_{532}$ values,
we separated the backscatter profiles in two components: the first arising from the contribution of the weakly depolarizing particles ($\delta_{nd} = 0.05$ for non-dust particles) and the second from the contribution of strongly depolarizing particles ($\delta_d = 0.31$ for dust particles). Then, the dust-related backscatter coefficient $\beta_d$ at 532 nm was obtained, following the procedure described by Tesche et al. (2009). The estimation of the height-resolved mass concentration (in  kg m$^{-3}$) of dust particles was based on the procedure described by Ansmann et al. (2012),
using the following equation

$$mass_d = \rho_d(v_d/\tau_d)\beta_d \, LR_d \quad (1)$$

where in our study the coarse-particle mass density ($\rho_d$) was assumed equal to 2.6 g m$^{-3}$ and a mean volume-to-AOT ratio for coarse mode particles, $v_d/\tau_d$ was calculated from AERONET measurements (https://aeronet.gsfc.nasa.gov/) for each station during the period 2014—2017. Table 2 summarizes these values for
the entire studied period, since only few cases were common in EARLINET and AERONET database. Regarding the LR$_d$ parameter, the mean LR values per station, as calculated from the lidar measurements, were used (cf. Table 2; Fig. 4c). These values are in good agreement with literature findings for long-range transported Saharan dust events (Tesche et al., 2009; Guerrero-Rascado et al., 2009; Ansmann et al., 2012; Groß et al., 2011; 2013; Bravo-Aranda et al., 2015).

Table 2: Assumed ($\rho_d$) and computed parameters ($v_d/\tau_d$, LR$_d$) used for the estimation of the height-resolved mass concentration (in kg m$^{-3}$) of dust particles. The ratio $v_d/\tau_d$ is derived from AERONET sun–sky photometer measurements within the period 2014—2017 at Granada, Potenza, Athens and Limassol. The LR$_d$ is calculated from the available corresponding lidar measurements per station.

| Station | $\rho_d$ (g m$^{-3}$) | $v_d/\tau_d$ (μm) (AERONET) | LR$_d$ (sr) |
|---------|------------------------|------------------------------|-------------|
| GRA | 2.6 | 0.80±0.29 | 52±8 |
| POT | 2.6 | 0.71±0.37 | 51±9 |
| ATZ | 2.6 | 0.94±0.50 | 52±9 |
| LIM | 2.6 | 0.87±0.27 | 49±6 |

### 3.2. The Spheroidal Inversion eXperiments (SphInX) software tool

The SphInX software provides an automated process to carry out calculations from lidar data to obtain the aerosol microphysical properties and further to statistically evaluate the inversion outcomes. It has been developed at the University of Potsdam (Samaras, 2016) within the Initial Training for atmospheric Remote Sensing (ITaRS) project (2012–2016). SphInX operates with expendable pre-calculated discretization databases based on spline collocation and on look-up tables of scattering efficiencies using T-matrix theory (Rother and Kahnert, 2009). This is to avoid the computational cost which would otherwise limit the microphysical retrieval to an impractical point. The Complex Refractive Index (CRI) is fed to the software separately for the real and imaginary parts which then constitutes a grid combining the following default values: Real part (RRI) $[1.33, 1.4, 1.5, 1.6, 1.7, 1.8]$ and Imaginary part (IRI) $[0, 0.001, 0.005, 0.01, 0.03, 0.05, 0.1]$. A range of values for the effective radius ($R_{eff}$), which occurs from the ratio of the total volume concentration ($u_t$) and the total surface-area concentration($a_t$), $r_{eff} = 3\,{u_t}/{a_t}$, is also needed to be predefined. The methodology applied here for spheroid-particle approximation is the same as presented in Soupiona et al. (2019). More specifically, the Raman lidar profiles were used as inputs for specific heights within the observed dusty layers and were averaged to produce the 6-point dataset of the so-called $3\beta_{par} + 2\alpha_{par} + 1\delta$ setup. All cases fulfilling this setup were treated in parallel for retrieving their microphysical properties. Here, the $R_{eff}$ ranged between 0.01 μm and 2.2 μm and the CRI grid was narrowed down to include only the values 1.4 and 1.5 for the Real part (RRI), and the values 0, 0.001, 0.005, and 0.01 for the Imaginary part (IRI), providing a total of 8 possible combinations for the CRI grid (instead of the initial total of 42 CRI grid). These ranges were used after a careful investigation of the values of the aerosol optical and microphysical properties found in the literature concerning transported Saharan dust events (Dubovik et al., 2006; Weinzierl et al., 2011; Mishra et al., 2014; Veselovskii et al., 2016; Benavent-Oltra et al., 2017; Veselovskii et al., 2020) in order to avoid retrieving less realistic dust-related size distributions and CRI values and to minimize the computational time. The outputs presented here are the RRI and IRI, the Single Scattering Albedo (SSA) and the $R_{eff}$.

### 3.3. Atmospheric dust cycle model (BCS-DREAM8b)

The BSC-DREAM8b model (Basart et al., 2012), operated by the Barcelona Supercomputer Center (BSC-CNS: www.bsc.es) provides operational forecasts since May 2009. The BSC-DREAM8b is a regional model designed to simulate and predict the atmospheric cycle of mineral dust aerosols. It is one of the most widely used and evaluated models for dust studies over northern Africa and Europe (cf. Jiménez-Guerrero et al., 2008; Papayannis et al., 2009; Basart et al., 2012; Amiridis et al., 2013; Tsekeri et al., 2017). The presented analysis includes vertical profiles of dust mass concentration simulations (0.3º x 0.3º horizontal resolution, 24 vertical levels, (from ground level to approximately 15 km in the vertical) corresponding to the studied cases and for time periods close to the measurement times, usually at 18:00 and 00:00 UTC, since the meteorological fields are initialized every 24 h (at 12:00 UTC) with the National Centers for Environmental Prediction (NCEP) global analysis (0.5ºx0.5º) and the boundary conditions are updated every 6 h with the NCEP Global Forecast System (GFS) (0.5ºx0.5º).

### 3.4. Radiative forcing simulations

The aerosol effects on solar and terrestrial radiation are usually quantified through the so-called aerosol radiative forcing (ARF). The aerosol radiative forcing (ARF) defined here as the perturbation in flux in the atmosphere caused by the presence of the dusty layers in relation to that calculated under clear sky conditions, can be expressed as (Quijano et al., 2000; Sicard et al., 2014; Mishra et al., 2014):

$$ARF(z) = \Delta F^{dusty}(z) - \Delta F^{clear}(z) \quad (2)$$

where, the net flux, $\Delta F$ at a level z, is the difference between the downwelling, and upwelling flux, $F\downarrow$ and $F\uparrow$, respectively:

$$\Delta F(z) = F \downarrow (z) - F \uparrow (z) \quad (3)$$

These fluxes (in W m$^{-2}$) are calculated separately for SW and LW radiation sources and assuming that the amount of the incoming solar radiation at the TOA is equal for both cases with and without the presence of dust aerosols. Therefore, the net ARF, $ARF_{NET}(z)$, is expressed as:

$$ARF_{NET}(z) = ARF_{SW}(z) + ARF_{LW}(z). \quad (4)$$

Based on this definition, the ARF at a given altitude will be positive when the aerosols cause a heating effect and negative when they cause a cooling effect. Finally, the ARF within the atmosphere ($ARF_{Atm}$) can be defined as the net difference between ARF at the top of the atmosphere (TOA) and the bottom of the atmosphere (BOA), denoted here as $ARF_{TOA}$ and $ARF_{BOA}$, respectively:

$$ARF_{Atm} = ARF_{TOA} - ARF_{BOA} \quad (5)$$

Our analysis was based on these equations for estimating the radiative forcing by means of direct and diffuse irradiances of an accurate radiative transfer model combining lidar measurements and dust concentration simulations.

### 3.4.1. The radiative transfer model (LibRadtran)

In this study, the downwelling and upwelling shortwave (280—2500 nm) and longwave (2.5—40 µm) irradiances at TOA and BOA levels have been simulated with the LibRadtran radiative transfer model version 2.0.2. (Emde et al., 2016). This software package contains numerous tools to perform various aspects of atmospheric radiative transfer calculations. In our study, the *uvspec* program that calculates the radiation field in the Earth's atmosphere was implemented for the *disort* radiative transfer equation (1-D geometry). Mid-latitude conditions for a typical Air Force Geophysics Laboratory (AFGL Atmospheric Constituent Profiles, 0–120 km, Anderson et al., 1986) and a typical surface albedo value (0.16) for urban areas (Dhakal, 2002) in the SW range were taken into account based also on visual observations. The OPAC library 4.0 (Koepke et al., 2015) was used for desert spheroids (T-matrix calculations) to determine aerosols' radiative properties in the aforementioned wavelength ranges. The non-spherical approximation is given by typical particle size dependent aspect ratio distributions of spheroids, derived from measurements at observation campaigns. In our study, the mineral particles of each case were treated by the model as spheroids for the mineral accumulation mode (MIAM) with $R_{MIAM} \in [0.005, 20]$ in µm.

A set of four simulations was carried out per case of the studied dust events. The first two simulations refer to clear-sky atmospheres with background/baseline aerosol conditions (default properties: rural type aerosol in the boundary layer, background aerosol above 2 km, spring-summer conditions and a visibility of 50 km, index "clear" in Eq. 2), the first for the SW and the second for LW range, since these ranges are treated separately by LibRadtran. The remaining two simulations correspond to dust loaded atmosphere, again, the one for the SW range and the other for the LW range, respectively, for which the vertical profiles of the dusty layers were used as additional inputs (index "dusty" in Eq. 2). These inputs have been obtained by three different Schemes: A) vertical mass concentration profiles simulated by the BCS-DREAM8b model, B) vertical mass concentration profiles of only the dust component as calculated from Eq. 1 (mass$_d$) utilizing the $\beta_{532}$ coefficient and, C) vertical profiles of $\alpha_{532}$ along with the respective mean AOT$_{532}$ value. In the final step, we calculated the parameters $\Delta F$, ARF, ARF$_{NET}$ and ARF$_{ATM}$ applying Eqs. 2—5.

The flowchart in Fig. 2 depicts these three Schemes applied to create the input files for the dust-loaded atmospheric conditions used in LibRadtran software package (Emde et al., 2016). Scheme A refers to the dust mass concentration as estimated by BSC-DREAM8b over the studied sites. I*n* Scheme B, only the dust vertical distribution is used as input, (based on the separation of the $\beta_{532}$ into dust and non-dust components that led to the calculation of the vertical distribution of the dust-only mass concentration, Eq. 1) in order to determine the Dust Radiative Forcing (DRF). On the other hand, in Scheme C both contributions of dust and non-dust aerosols (total

αaer) are taken into account. Additionally, for Scheme C conversion factors from OPAC were used in order to convert the $\alpha_{aer}$ and the corresponding AOT from 532 nm to 10 μm (peak, within the atmospheric window). The conversion was based on an adaptive inversion algorithm of Shang et al. (2018) who presented a way to convert extinction coefficients at different wavelengths by using Ångström exponent values derived from AOTs. It should be mentioned here that the Scheme B, even though it also includes some assumptions and uncertainties in its calculations, is the only one, compared to the rest two (Schemes A and C) that gives us the opportunity to calculate only the dust contribution in the radiative effect.

For all these Schemes in this study, 30 vertical levels have been used between ground and 120 km height with a spatial vertical resolution of 0.5 km starting from ground level (BOA) to 2 km and from 5 to 10 km, a resolution of 0.2 km from 2 to 5 km, due to the presence of the dust layers within this height range and additionally at the heights of 20 and 120 km (TOA). All simulations were performed for three different Solar Zenith Angles (SZA), 25°, 45° and 65° covering a typical diurnal spring-summer cycle for radiative forcing estimates at mid-latitudes. For the very few available winter-time measurements that SZA does not reach 25°, a theoretical approach on the ARF is estimated. All cases were treated for cloud-free conditions. Except the altitude in km ($z_{out}$), the additional outputs that have been implemented in our Schemes are: the direct irradiance ($e_{dir}$), the global irradiance ($e_{glo}$) the diffuse downward irradiance ($e_{dn}$), the diffuse upward irradiance ($e_{up}$), and the heating rates (heat) in K day$^{-1}$, as described by Mayer et al. (2017).

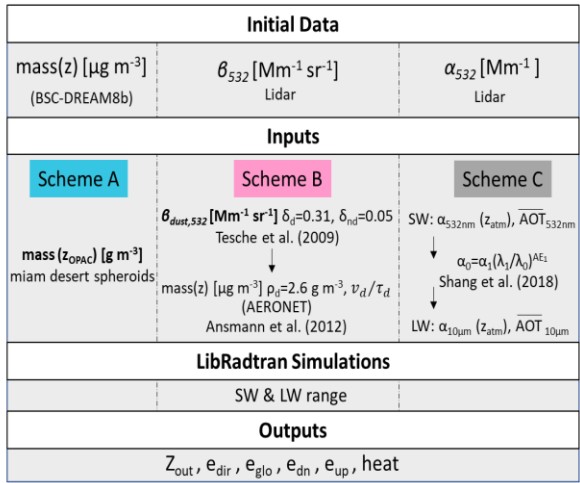

Figure 2: Flowchart of the three Schemes used to retrieve simulations of irradiances using the LibRadtran software package.

### 3.4.2. Radiation data set

LibRadtran irradiance outputs have been validated against reference solar irradiance pyranometer measurements at the Earth's surface (Kosmopoulos et al., 2018). For this study, solar radiation data measured by pyranometers were available only for the Granada and Athens stations. The evaluation was performed using cloudless time periods only. The reference solar radiation data set consists of one-minute simultaneous measurements of horizontal global and diffuse irradiance measured with two CMP11 pyranometers, at Granada, and two CMP21, at Athens (located at National Observatory of Athens actinometric station in the Penteli area, 10 km from NTUA). These pyranometer models, both manufactured by Kipp & Zonen, have a black-coated thermopile acting as a sensor which is protected against the meteorological conditions by two concentric hemi-spherical domes. They both comply with the International Organization for Standardization (ISO) 9060 (1990) criteria for an ISO secondary standard pyranometer, being classified as "high quality" according to the World Meteorological Organization (WMO) nomenclature (WMO, 2018). Additionally, the corresponding pyranometer measuring the diffuse component was mounted on a shading device to block the direct irradiance and prevent it from reaching the

sensor. In this study, the shading devices employed were a Solys2 sun tracker and a CM121 shadow ring, at Granada and Athens respectively, both manufactured by Kipp & Zonen. For those diffuse irradiance measurements taken using a shadow ring, the model proposed by Drummond (1956) has been applied in order to correct for the diffuse radiation intercepted by the ring, as suggested by the manufacturer (Kipp & Zonen, 2004).

### 3.5. Evaluation of aerosol mass concentration vertical profiles

Before using the vertical dust mass concentrations profiles retrieved from i) BSC-DREAM8b model simulations (Scheme A) and ii) lidar measurements as calculated from Eq. 1 ($mass_d$), (Scheme B) as inputs to the LibRadtran model, we performed a day-by-day comparison between them. Due to the different spatial and vertical resolution between the modeled and the lidar profiles, both profiles were degraded to the fixed height levels of the OPAC dataset (0, 1, 2, 3, 4, 5, 6, 7, 8, 9, 10, 11, 12, 35 km).

Figure 3 shows the Taylor's diagram of the mass concentration simulated by the BSC-DREAM8b model against the lidar-retrieved ones. The azimuthal angle presents the correlation coefficient, the radial distance presents the normalized Standard Deviation (SD) of each point, the root mean square error (RMSE) is proportional to the distance from the point on the x-axis identified as "Calculated", while the latter, is depicted by a black point at the (1,0) cross section, indicates the lidar retrieved aerosol mass values representing the reference point. The normalization of the SD is performed with respect to the calculated values. In the 66 % of the cases there is a good correlation ($r > 0.6$), and consequently, a good prediction of the shape of the vertical distribution is achieved, while in 96 % of the cases the model gives lower concentration values (*normalized SD* $< 1$) revealing an underestimation in the intensity and the mass concentration of the events. Therefore, we report a mean underestimation of the mean mass concentration values of the BSC-DREAM8b of the order of 31%. However, we should take into consideration: i) the spatial resolution, where the lidar observations are considered as point measurements while the simulations represent uniform pixels of 0.3° resolution and ii) the temporal resolution, where the lidar retrieved profiles are hourly averaged, while the model derived profiles are instantaneous results, saved every 6 hours.

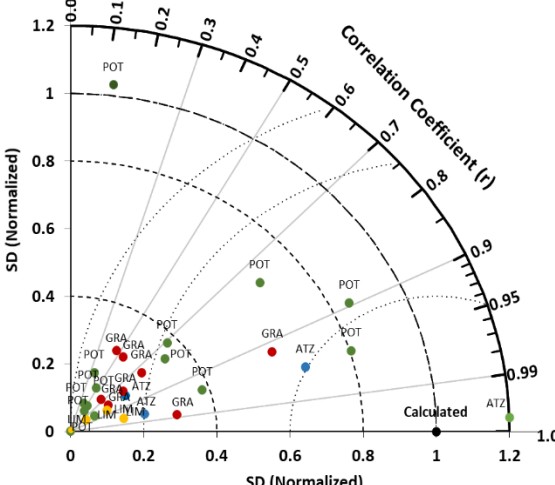

Figure 3: Taylor's diagram of the case-by-case vertical mass concentration simulated by BSC-DREAM8b model against the lidar retrieved ones. The black point (1,0) represent the calculated lidar data. The azimuthal angle presents the correlation coefficient (r), the radial distance of any point from the origin (0,0) indicates the normalized SD of the data set. The normalization of the SD is performed with respect to the calculated values. The colored the dots represent each one of the 4 EARLINET stations, namely GRA (red), POT (green), ATZ (blue) and LIM (orange).

By further comparing the modeled mass vertical profiles to the ones calculated by lidar, we report that the mean center of mass (in km) estimated from BSC-DREAM8b profiles is 0.6 km lower than the one calculated from the lidar measurements ($2.6 \pm 1.0$ km and $3.2 \pm 1.1$ km respectively). The maximum concentration (peak) is

usually found in the region 2-3 km, both in the modeled and the observed data. The BSC-DREAM8b, having a significantly lower vertical resolution compared to the lidar, predicts smoother profiles of dust layers by spreading the layer's base to lower altitudes (~1km, in 100% of the cases) and the top at higher altitudes (in 86% of the cases) compared to the observed ones. These remarks are in line with the previous studies of Mona et al. (2014) and Binietoglou et al. (2015) where they have reported discrepancies concerning the base, the top layer height and extinction profiles and good agreement in terms of profile shape, between the BSC-DREAM8b and observations. However, since fixed height levels of the OPAC dataset were finally used in LibRadtran for the ARF simulations of the three Schemes, having significantly lower vertical resolution compared to the intitial lidar profiles, these discrepancies in height were smoothed out.

### 3.6. Evaluation of ground level LibRadtran outputs

The evaluation of the performance of the model was undertaken by statistical means. The relative root mean square error (rRMSE), the relative mean bias error (rMBE), the correlation coefficient (r) and the normalized SD were calculated in order to numerically quantify the performance of the global irradiance recorded by pyranometers and simulated from the three Schemes. Table 3 shows the statistical results for the modeled global irradiance values versus the reference one-minute pyranometer measurements for both locations (Granada, Athens) and the three Schemes at 25°, 45° and 65° SZA. All Scheme simulations perform remarkably well, with rRMSE values ranging from 8.3 to 16.2% and rMBE values between 0 and 15.2%. In general, the rRMSE is slightly higher at Granada, mainly for Scheme A. According to this statistic, the LibRadtran outputs with the best performance are those obtained by Scheme C as input followed by Schemes B and A, respectively. This order is the same attending to the rMBE values with the exception of Scheme A at Athens. The correlation coefficient r depicts the good performance of the radiative transfer model for the three Schemes and the two locations. All simulations present a value of $r >$ 0.95 with minor differences (below a 10%) in the normalized SD values respect to the pyranometer global irradiance values. A slight overestimation is observed for all scheme outputs at Granada (norm SD > 1). Conversely, this overestimation is no longer evident in the modeled global irradiance for Athens. However, it is important to note the good performance of the Scheme B despite the high number of various parameters involved in it.

Table 3: Statistical metrics for the modeled global irradiance values versus the reference pyranometer measurements for Granada and Athens and the threes Schemes applied.

|  | Granada | | | | Athens | | | |
|---|---|---|---|---|---|---|---|---|
|  | rRMSE (%) | rMBE (%) | r | SD (norm) | rRMSE (%) | rMBE (%) | r | SD (norm) |
| **Scheme A** | 16.2 | 15.2 | 0.99 | 1.09 | 10.8 | - 0.2 | 0.97 | 0.89 |
| **Scheme B** | 11.9 | 5.7 | 0.97 | 1.10 | 10.2 | 8.3 | 0.99 | 0.92 |
| **Scheme C** | 8.8 | 5.9 | 0.99 | 1.09 | 8.3 | 6.3 | 0.99 | 0.96 |

## 4.     Results
### 4.1. Aerosol geometrical and optical properties per site

For each case studied, the mean $\delta_{p532}$, mean $LR_{532}$ and $AOT_{532}$ values were calculated inside the dust layers (cf. Sect.2.2) as shown in Fig. 4 (a—d). The corresponding SD values give an indication of the variability of these parameters from base to the top of the dust layer. Figure 4a shows the aerosol geometrical properties for the detected layers one by one, per station and per year. The mean values of the base and top height of the dust layers per station, along with their SD are marked with horizontal bounded lines. At the four sites (Granada, Potenza, Athens and Limassol) mean layer thicknesses of $3392 \pm 1458$ m, $2150 \pm 1082$ m, $1872 \pm 816$ m and $1716 \pm 567$ m were calculated respectively. We also found mean $\delta_{p532}$ values of $0.24 \pm 0.05$, $0.26 \pm 0.06$, $0.28 \pm 0.05$ and $0.28 \pm 0.04$ (Fig. 4b), and indicative mean $AOT_{532}$ values of $0.40 \pm 0.31$, $0.11 \pm 0.07$, $0.12 \pm 0.10$ and $0.32 \pm 0.17$ (Fig. 4d), respectively. Similar mean $LR_{532}$ values of around 51 sr (Fig. 4c) were found for all stations. The Granada

station holds the minimum mean value for layers' base height ($1567 \pm 788$ m a. s. l.) and the maximum for top height ($4960 \pm 975$ m) and layer's thickness. Concerning the LR values no remarkable deviations were observed among the four stations having mean values around 51 sr, which are in very good agreement with literature findings (Tesche et al., 2009; Ansmann et al., 2012; Groß et al., 2011; 2013). The largest indicative mean AOT value, equal to $0.40 \pm 0.31$, observed over Granada station for the total studied period is in accordance to the geometrical properties (Fig. 4a) that depict thick dust layers in the majority of the cases.

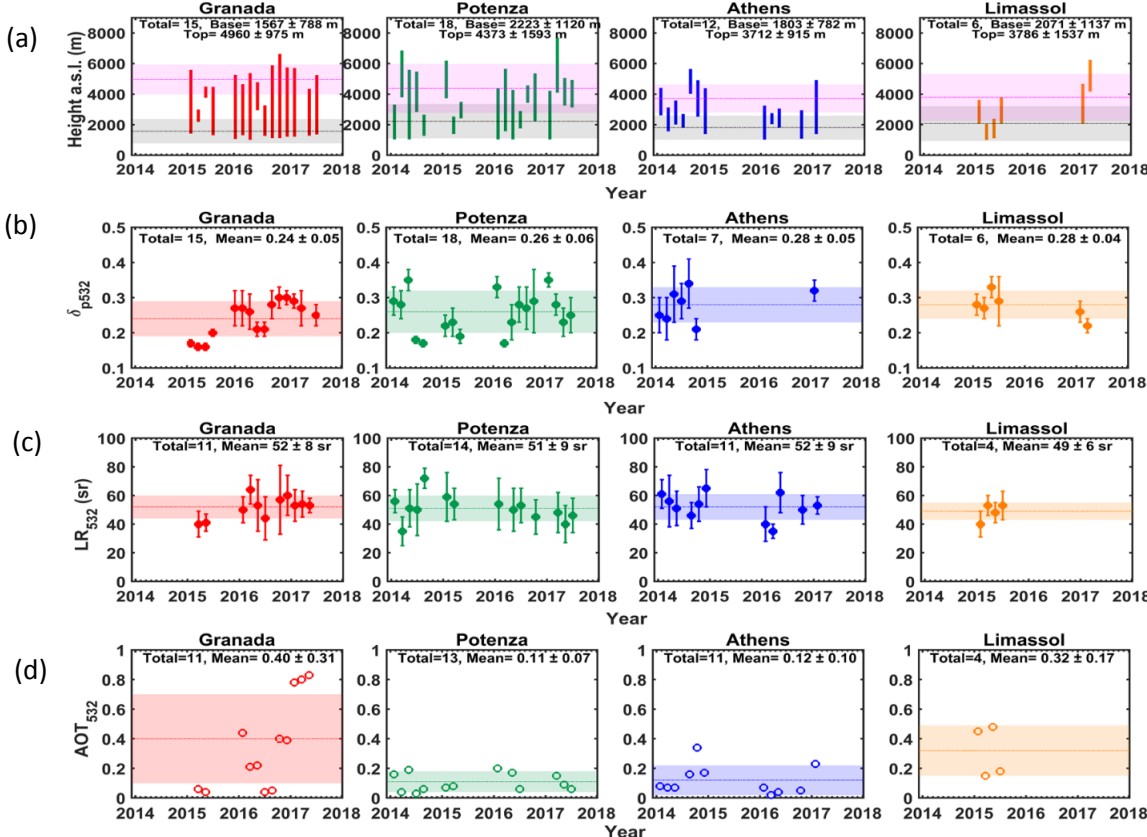

Figure 4: Mean values along with standard deviation of a) base and top, b) $\delta_{p532}$, c) $LR_{532}$ and d) $AOT_{532}$, per station (text and banded lines) and per case (symbols and error bars) within the observed dust layers.

Considering Granada's station as representative of the west Mediterranean region, Potenza of the central Mediterranean region, Athens and Limassol stations of the Eastern Mediterranean region, a dust aerosol mode classification per region can be made. For this purpose, the mean $AOT_{532}$ versus the $AE_{\beta532/1064}$ giving an indication of the aerosol particle size in the atmospheric column for each region are shown in Fig. 5. A wide spread of the AOT values at moderate to low $AE_{\beta532/1064}$ values (between 0 and 0.6) observed in the Western Mediterranean region, demonstrates that the dust size distribution in this area is dominated by coarse mode particles during events of different intensities. On the other hand, the presence of dusty layers, in the central and Eastern Mediterranean regions can be associated with higher $AE_{\beta532/1064}$ values (even up to 1.5) and consequently, with the presence of fine mode particles and lower dust loads. Our findings verify that the longer the time/distance of dust transport is, the more likely is for the dust aerosols to be mixed with background ones in the Eastern Mediterranean (Groß et al., 2019; Soupiona et al., 2019).

In terms of the aerosol size distribution, the scatter plot of Fig. 5 allowed to perform a k-means clustering (Arthur and Vassilvitskii, 2007) in order to define three physically interpretable aerosol size distributions: a) fine mode, with $AE_{\beta532/1064} > 0.6$, b) coarse mode, with $AE_{\beta532/1064} \leq 0.6$ and $AOT_{532}$ between 0 and 0.2, c) whilst

$AE_{\beta532/1064}$ values smaller than 0.6 attributed to large AOTs (between 0.2 and 0.8) are representative of extreme dust events. It seems that the majority of these extreme dust outbreaks occur over the Western Mediterranean region, more likely due to its location close to the African continent. For example, Fernández et al. (2019) recently reported an unprecedented extreme winter time Saharan dust event, during February 2017, over the whole Iberian Peninsula with AOTs > 0.2 (675 nm) and $AE$ values around zero. More studies referring to the occurrence of extreme dust
events over the aforementioned area can be found in literature (Cachorro et al., 2008; Guerrero-Rascado et al., 2009).

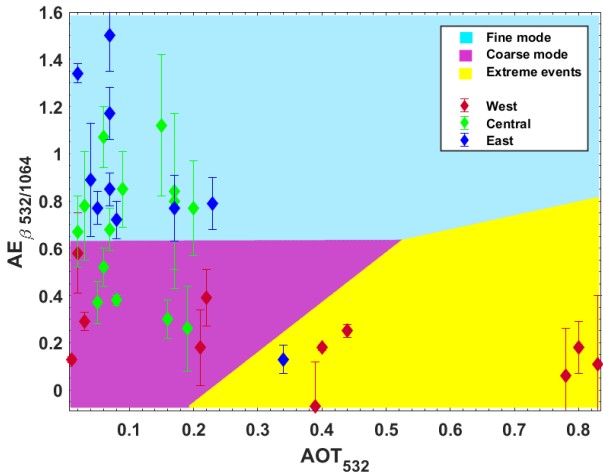

Figure 5: $AE_{\beta532/1064}$ versus $AOT_{532}$ per region: west (red marks and error bars), central (green marks and error bars) and east (blue marks and error bars) Mediterranean region. K-means clustering revealed three clusters: fine mode (light blue
background), coarse mode (magenda background) and extreme dust events (yellow background).

## 4.2. Clustering per mixing state

Based on the High Spectral Resolution Lidar (HSRL) classification presented by Groß et al. (2013), the intensive aerosol quantities $LR_{532}$ versus $\delta_{p532}$ were plotted, identifying three of the six existing clusters in our data (Fig. 6). The first cluster (green marks and error bars) represents a mixing state of Saharan dust and BB aerosols
having a large spreading in mean LR values and low mean $\delta_{p532}$ values (40 sr $\leq LR_{532} \leq$ 75 sr, 0.16 $\leq \delta_{p532} \leq$ 0.19). The second one, (red marks and error bars) is attributed to mixed Saharan dust, where dust aerosols are dominant, but urban/continental, marine or even pollen aerosols are also possibly present (40 sr $\leq LR_{532} \leq$65 sr, 0.20 $\leq \delta_{p532} \leq$ 0.29). The third cluster (orange marks and error bars) is attributed to pure Saharan dust aerosols (45 sr $\leq LR_{532} \leq$ 60 sr, 0.30 $\leq \delta_{p532} \leq$ 0.36). The most populated and consequently, the
most common, among those three clusters is the red one, as expected, due to the frequent mixing of dust aerosols with continental ones (Papayannis et al, 2008). The range of our measured $\delta_{p532}$ values as indicated with the horizontal error bars in Fig. 6, overlap between the three identified aerosol clusters, showing a more realistic transition from one cluster to the other, bridging the gap especially between green and red clusters from the HSRL classification.

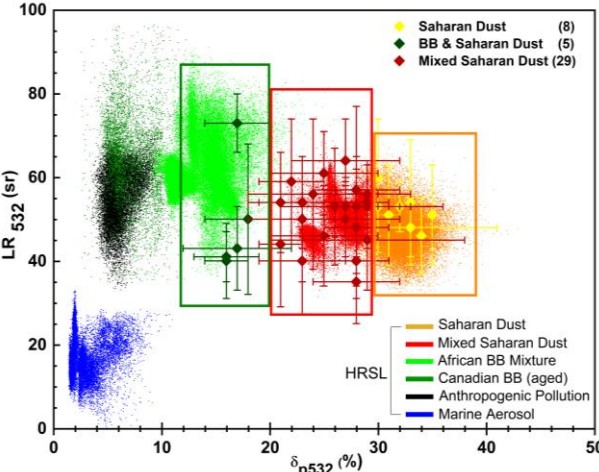


Figure 6: $LR_{532}$ versus $\delta_{p532}$ values from HSRL observations presented by Groß et al. (2013), (colored dots) along with the selected datasets from the four EARLINET stations (symbols and error bars).

Table 4 summarizes the mean values of the aerosol geometrical, optical, and microphysical properties of the three identified clusters along with their SD (5 cases for BB & Saharan dust, 8 cases for Saharan dust, 29 cases for
mixed Saharan dust). A synergistic approach of HYSPLIT (trajectories of 120 hours backward for each case) and Google Earth (distance calculator) tool allowed us to estimate the distance travelled (in km) to the respective sites and the mixing hours per cluster. Specifically, the term of mixing refers to the hours the air masses travelled after leaving the African continent. We can see that the Saharan dust cluster presents the lowest mixing with other air masses ($26 \pm 13$ hours), compared to the other clusters ($41 \pm 26$ hours for the BB & mixtures and $31 \pm 13$ hours
for the mixed Saharan dust cluster). Moreover, the air masses of the Saharan dust cluster seem to travel faster than those of the other two clusters, although covering a greater distance ($4845 \pm 2825$ km) at the same time (within 120 hours). Now, the main difference between the two remaining clusters (BB & mixtures/Mixed Saharan Dust) is attributed to the mixing hours. The air masses of the latter cluster remain 15 hours longer and circulate over the Mediterranean and Europe, so they are probably enriched with other types of aerosols.

Concerning the aerosol optical properties, the $\beta_{532}$ and $\alpha_{532}$ show lower values for BB & dust and for mixed Saharan dust cases ($1.10 \pm 0.15$ x $10^{-3}$ km$^{-1}$ sr$^{-1}$, $0.47 \pm 0.28$ km$^{-1}$ and $1.24 \pm 0.80$ x $10^{-3}$ km$^{-1}$ sr$^{-1}$, $0.74 \pm 0.48$ km$^{-1}$ respectively) and higher values ($1.54 \pm 1.05$ x $10^{-3}$ km$^{-1}$sr$^{-1}$, $0.80 \pm 0.27$ km$^{-1}$) for the Saharan dust cluster. Therefore, higher AOT$_{532}$ values ($0.32 \pm 0.25$) were found for the latter cluster compared to the others, due to the higher dust burden of these events over the affected sites. The highest $\delta_{p532}$ values ($0.32 \pm 0.02$), indicate
the arid origin and the coarse mode of pure Saharan dust layers (Freudenthaler et al., 2009) of the corresponding cluster. No direct information can be extracted from the similar $LR_{532}$ values about the mixing state of the aerosol layer, except that the range of the SD narrows as the mixing decreases. However, for the cases that observations at 355 nm were available, it seems that the lidar ratio color ratio (namely the $LR_{355}/ LR_{532}$) converges to unity for the Saharan dust cluster, indicating the absence of spectral dependence for the case of pure dust (Müller et al., 2007;
Veselovskii et al., 2020). For these cases also, the AE$_{\beta355/532}$ becomes closer to zero taking mean value of $0.35 \pm$ 0.45.

We also summarise the changes in mean microphysical properties estimated with SphInX tool for all the cases of each of the three identified clusters. The BB & Saharan dust cluster takes the lower mean R$_{eff}$ value ($0.293 \pm 0.074$ μm) due to the fine structure of BB aerosols included in the layer, while a mean R$_{eff}$ of $0.360 \pm$
$0.081$ μm corresponds to the cluster of mixed Saharan dust and a slightly larger value ($0.387 \pm 0.070$ μm) corresponds to the Saharan dust cluster. The values for RRI, IRI and SSA at 532 nm were similar for the two clusters that not include BB aerosols, whilst the presence of BB aerosols of the first cluster leads to higher RRI and IRI

values and lower SSA, results that are in good agreement with the ones reported in Petzold et al. (2011) over Dakar, for mineral dust and dust mixed with anthropogenic pollution.

Table 4: Mean values of optical, geometrical and microphysical properties of the three identified clusters along with their standard deviation (SD). Zero SD indicates no variability in the corresponding retrieved parameter. The term of mixing refers to the hours the air masses travelled after leaving the African continent.

| Parameters | | Clusters | | |
|---|---|---|---|---|
| | | BB & Saharan Dust | Mixed Saharan Dust | Saharan Dust |
| **Optical Properties** | $\beta_{532}$ (km$^{-1}$sr$^{-1}$) | $1.10\pm0.15$ [x10$^{-3}$] | $1.24\pm0.80$ [x10$^{-3}$] | $1.54\pm1.05$ [x10$^{-3}$] |
| | $\alpha_{532}$ (km$^{-1}$) | $0.47\pm0.28$ | $0.74\pm0.48$ | $0.80\pm0.27$ |
| | LR$_{532}$ (sr) | $51\pm15$ | $50\pm7$ | $52\pm5$ |
| | LR$_{355}$ (sr) | $35\pm13$ | $42\pm7$ | $51\pm10$ |
| | $\delta_{p532}$ | $0.17\pm0.01$ | $0.26\pm0.03$ | $0.32\pm0.02$ |
| | LR$_{355}$/LR$_{532}$ | $0.69\pm0.24$ | $0.84\pm0.16$ | $0.98\pm0.16$ |
| | AE$_{\beta355/532}$ | $0.44\pm0.59$ | $0.52\pm0.61$ | $0.35\pm0.45$ |
| | AOT$_{532}$ | $0.03\pm0.02$ | $0.15\pm0.10$ | $0.32\pm0.25$ |
| **Geometry & Mixing** | Thickness (km) | $0.79\pm0.21$ | $2.08\pm0.76$ | $3.10\pm1.72$ |
| | Distance (km) | $3496\pm1185$ | $3662\pm1617$ | $4845\pm2825$ |
| | Mixing (hours) | $41\pm26$ | $66\pm41$ | $26\pm13$ |
| **Microphysical Properties** | R$_{eff}$ ($\mu$m) | $0.293\pm0.074$ | $0.360\pm0.081$ | $0.387\pm0.070$ |
| | RRI | $1.50\pm0.00$ | $1.47\pm0.05$ | $1.47\pm0.05$ |
| | IRI | $0.005\pm0.000$ | $0.0046\pm0.0045$ | $0.0041\pm0.0018$ |
| | SSA$_{532}$ | $0.948\pm0.002$ | $0.964\pm0.018$ | $0.964\pm0.022$ |
| | SSA$_{355}$ | $0.937\pm0.007$ | $0.958\pm0.022$ | $0.952\pm0.026$ |

### 4.3. Regional aerosol radiative forcing (ARF)

As mentioned previously, there is shortage of papers in the literature about the role of dust on the Earth's radiation budget. Since very few in situ measurements of ARF effects and heat fluxes are available especially in the Mediterranean (Bauer et al., 2011; Meloni et al., 2018), we are restricted to perform simulations to quantify the role of dust aerosols on the radiative forcing in the studied regions. The mean ARF is calculated during this simulation, calling twice the LibRadtran radiation code: with (index "dusty" in Eq. 2) and without (index "clear" in Eq. 2) the 475  presence of free tropospheric dusty aerosol layers. For all cases, the vertical profiles of ARF starting from ground level/bottom of atmosphere (BOA) up to the top of atmosphere (TOA) in the SW and LW ranges were simulated using the three aforementioned Schemes.

A negative forcing of aerosols both at the BOA and TOA is noted in the SW range, as presented in Fig. 7a, which depicts the mean ARF of all cases per scheme, over the Mediterranean Basin. Our results indicate a presence 480  of less absorbing aerosols, thus having a cooling behavior. Depending on the dust optical properties and load intensity, ARF values at the BOA range from -40 to -13 W m$^{-2}$ at SZA 25$^o$, from -43 to -14 W m$^{-2}$ at SZA 45$^o$ and from -44 to -15 W m$^{-2}$ at 65$^o$. At the TOA, the corresponding ranges per SZA are -9.5 to -1.4 W m$^{-2}$ (25$^o$), -16 to -3.3 W m$^{-2}$ (45$^o$) and -24.3 to -6.9 W m$^{-2}$ (65$^o$). Similarly, in the SZA independent LW range (thermal spectral range), the ARF values range from +1.6 to +4.6 W m$^{-2}$ for the BOA and from +0.8 to +3.6 W m$^{-2}$ for the TOA. Our 485  estimations are consistent with results obtained by other literature findings for Saharan dust aerosols over the Mediterranean region. Specifically, Sicard et al. (2014) found that the SW RF at the BOA has always a cooling effect varying from -93.1 to -0.5 W m$^{-2}$ while the corresponding LW RF has always a heating effect varying from +2.8 to +10.2 W m$^{-2}$. They also concluded that dust aerosols have a cooling effect in the SW spectral range at the TOA with a RF ranging from -24.6 to -1.3 W m$^{-2}$ while at the TOA the LW RF varies between +0.6 and +5.8 W 490  m$^{-2}$. Meloni et al. (2003) found at the island of Lampedusa instantaneous RF of -70.8 W m$^{-2}$ at the BOA and

-7.5 W m$^{-2}$ at the TOA within the range 300-800 nm for an event with AOT of 0.51 at 415 nm. For the same location and for another strong Saharan dust outbreak (AOT$_{500}$=0.59), Meloni et al. (2015) reported a total (SW + LW) radiative forcing of  -48.9 W m$^{-2}$ at the BOA, -40.5 W m$^{-2}$ at TOA, and +8.4 W m$^{-2}$ within the atmosphere for SZA=55.1°. A negative radiative effect reaching down to -34.8 W m$^{-2}$ at the BOA in the Mediterranean area was also recently reported by Gkikas et al. (2018) for the studied period March 2000—February 2013.

Variations among these values are expected since they strongly depend on the different AOTs, mass estimations and extinction profiles. Estimations retrieved from Scheme B are expected to give higher values compared to those given from Scheme A as revealed also by Fig.3. The ARF at the LW spectral region is opposite in sign and significantly lower in absolute values than in the SW region. The difference between the TOA and BOA ARF, with the former to be only weakly perturbed and the latter to be quite stronger, can be attributed to the heating within the troposphere, since the presence of the dust aerosols mainly leads to a displace of surface's radiative heating into the dusty layer. We also noticed that the low values of the reflected solar flux are partially offset by the absorption of upwelling LW radiation. Finally, in the LW spectral region, the mean ARF values at the BOA (Scheme A: +1.6±1.6 W m$^{-2}$, Scheme B: +4.6±4.7 W m$^{-2}$, and Scheme C: +2.9±9.4 W m$^{-2}$) are close to those at the TOA (Scheme A: +0.8±0.9 W m$^{-2}$, Scheme B: +3.6±4.4 W m$^{-2}$, and Scheme C: +1.2±6.2 W m$^{-2}$) but moved a little to more positive values. As a result, the ARF$_{Atm}$ (Eq. 5) is positive during the diurnal circle, yielding net radiative heating of the dust layers.

The mean net heating rate within the atmosphere, calculated by adding algebraically both rates in the SW and LW spectral ranges is presented in Fig. 7b. Here, the net heating rate is clearly dependent on the available solar radiation and increases with SZA due to the low incoming solar radiation reaching the BOA at afternoon hours (SZA 65°). Our estimations are in accordance with the fact that as the SZA increases, the optical path of the SW radiation grows significantly, increasing the attenuation of the direct radiation while generating a higher fraction of the diffuse radiation. This effect is more pronounced at the BOA, in which, the intensity of the heating rate is reduced with increasing SZA, since fewer photons are available to heat the dust layers. The net heating rate values for Scheme A are: -0.05±0.04 K day$^{-1}$ (25°), -0.04±0.03 K day$^{-1}$ (45°) and 0.00±0.02 K day$^{-1}$ (65°). Similar to slightly higher values are observed for Scheme B as follow: -0.07±0.06 K day$^{-1}$ (25°), -0.04±0.03 K day$^{-1}$ (45°), and -0.02±0.02 K day$^{-1}$ (65°). For Scheme C, we report higher values of this parameter during the diurnal circle. More precisely, the net heating rate is almost 1.5 times higher at 25°, 2 times higher at 45° and around 0.8 times higher at 65°, compared to the aforementioned Schemes. Greater sensitivity in the SZA appears in Scheme B as it results from the line slope.

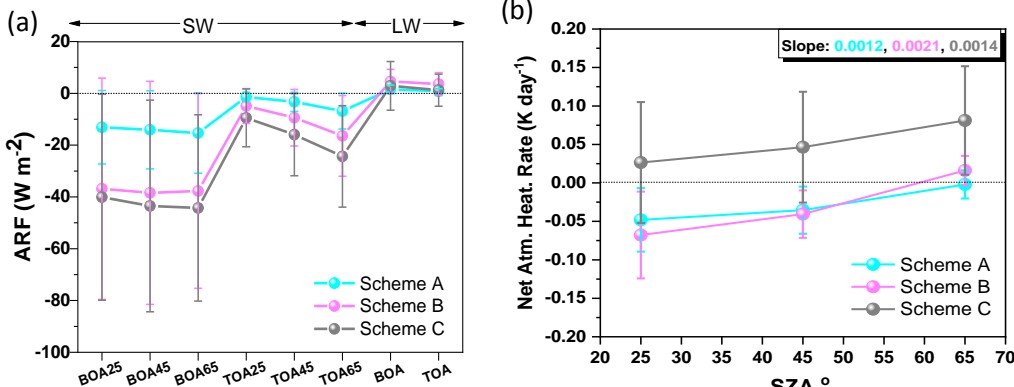

Figure 7: Mean values of a) SW and LW ARF at BOA and TOA and b) the net heating rate within the atmosphere, along with their SD for the three Schemes applied in the total set of the studied cases. The inserted box depicts the line slope.

In order to further explain the difference of sign in the net heating rate of Scheme C, compared to the two others presented in Fig. 7b, we plotted the aforementioned parameter along with the base layer height, the AOT$_{532}$

and the layer thickness of each case as presented in Fig. 8. Taking into account that the effect of net heating rate occurred by the dusty cases, from negative to positive values, is more pronounced close to surface at small SZA values, the estimations of the net heating rate at the BOA at 25° SZA were selected to be presented in this graph. It becomes clear that the sign of the net heating rate at BOA depends on the dust vertical structure and the AOT. More

precisely, the majority of the cases having low $AOT_{532}$ values ($\leq 0.2$) and low layer thickness ($\leq 2$ km) give negative net heating rate values. Additionally, the higher the AOT is, the higher the absolute value of the net heating rate is. Concerning the base layer height, it plays a key role to the absolute net heating rate of each case, since dust layers close to the ground take higher absolute net heating rate values. For example, let us examine two dust events occurred during the same month (August). The first one with base of 2.8 km, 0.73 km thickness and $AOT_{532}$ equal

to 0.01 has a heating rate of -0.17 K day$^{-1}$, while the second with base of 3.8 km, 0.66 km thickness and $AOT_{532}$ equal to 0.02 has a net heating rate of almost zero (-0.03 K day$^{-1}$). In another comparison, net heating rate values of -0.02 K day$^{-1}$ versus +0.09 K day$^{-1}$ are reported for two layers, during summer time that have almost the same base (2.6 km and 2.5 km) and thickness (2.3 km and 2.4 km) but different $AOT_{532}$ values (0.08 and 0.34 respectively). Finally, a combination of high $AOT_{532}$ ($0.21 - 0.83$) and high thickness ($2.1 - 5.5$ km), along with low base height

($1.0 - 1.5$ km), give high net heating rate values with positive sign ranging from +0.06 to +0.26 K day$^{-1}$.

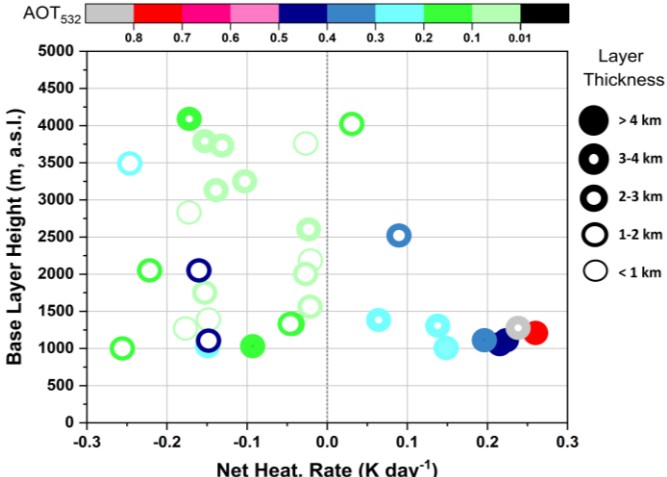

Figure 8: Net heating rate values per case of Scheme C estimated at BOA, 25° SZA versus base layer height. The horizontal colorbar indicates the $AOT_{532}$ values and the vertical symbol thickness indicates the layer thickness.

       Figures 9 (a-c) depicts the same results as in Fig. 7a but for each of the three identified clusters: BB and dust,

mixed Saharan dust and Saharan dust. The ARF in the SW range is negative both in the BOA and TOA for all clusters and is dominated by large dust particles for the cluster of the Saharan dust episodes (cf, Table 4; Fig. 9c), (Scheme A: -22.5±16.7 W m$^{-2}$, Scheme B: -38.3±29.0 W m$^{-2}$, Scheme C: -49.2±50.9 m$^{-2}$ for BOA, and Scheme A: -2.5±2.1 W m$^{-2}$, Scheme B: -4.9±4.1 W m$^{-2}$, Scheme C: -12.1±14.4 m$^{-2}$ for TOA, SZA 25°), whereas dust layers mixed with biomass burning aerosols have significantly lower cooling effect (Fig. 9a, Scheme A: -6.2±4.0 W m$^{-2}$,

Scheme B: -18.2±11.3 W m$^{-2}$, Scheme C: -4.8±3.5 W m$^{-2}$ for BOA, and Scheme A: -0.5±0.4 W m$^{-2}$, Scheme B: -2.0±1.6 W m$^{-2}$, Scheme C: -0.7±0.5 m$^{-2}$ for TOA, SZA 25°) for each of the three applied Schemes. ARF seems to be inversely proportional to the mixing ratio, since higher absolute values are estimated for less mixed dust layers. This can directly be linked to the fact that ARF values strongly depend on $\alpha_{par}$, $\beta_{par}$ and AOT that take much higher values for the Saharan dust cluster as already reported (Table 4). Focusing on the SW range, the cooling effect for

Scheme A of the Saharan dust cluster is up to 3 times higher compared to the BB and Saharan dust one, whilst the cooling effect for Scheme C of the former cluster is up to 10 times higher compared to the latter. The cooling effect of Scheme B becomes also stronger with the decreasing mixing state but in a lower magnitude (the former cluster is almost 2 times higher compared to the latter).

Hence, even though the studied cases included in the Saharan dust cluster usually take higher mass concentration values than the other cases, as predicted by BSC-DREAM8b (Scheme A), the model still seemingly underestimates the intensity of strong transported dust episodes over the observation stations. On the contrary, Scheme C is the most sensitive to the mixing state of the aerosol layers. To explain this result one should consider that on the one hand, spheroidal particles such as dust have larger dimensions than spherical ones such as BB aerosols and thus, lead to larger AOTs (Haapanala et al., 2012) and consequently to increased negative ARF and on the other hand, the Schemes A and B involve greater assumptions concerning dust particles than Scheme C.

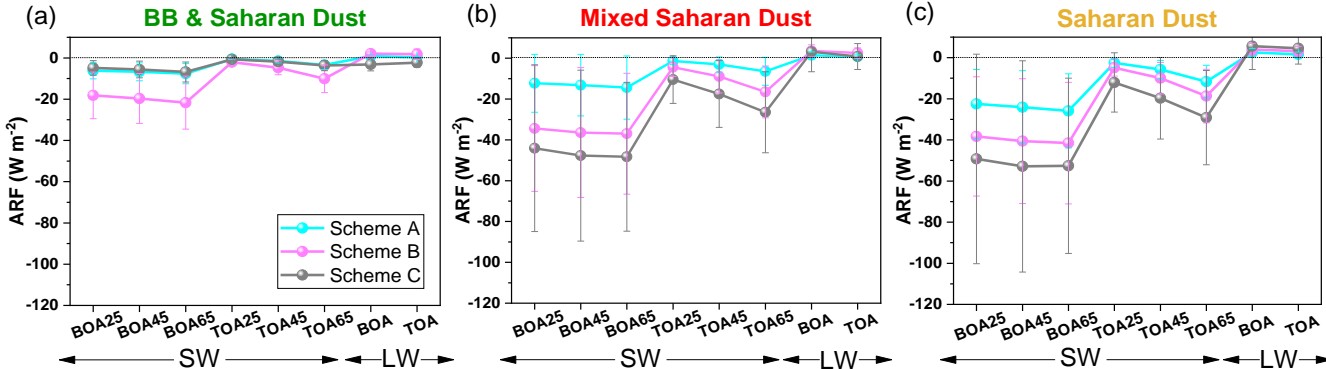

Figure 9: Mean values of SW and LW ARF at BOA and TOA and along with their SD for the three Schemes applied regarding the mixing state, namely a) BB & Saharan dust, b) Mixed Saharan dust and c) Saharan Dust. The dotted line represents the ARF zero value.

Finally, our interest is focused on the vertical ARF profiles from the surface (a.s.l.) up to 10 km height in the free troposphere, where airborne dust is usually found, as estimated by Scheme C at 45° SZA per station. The ARF profiles, in the SW region, presented in Figs. 10 (a-d), follow the aerosol extinction vertical structure. The ARF values at the BOA are high in absolute values with a cooling behavior and decreases with increasing height, while the magnitude is proportional to the aerosol load in the whole atmospheric column. Specifically, the ARF ranges from -150.0 to -1.9 W m$^{-2}$ for Granada, from -38.1 to -3.7 W m$^{-2}$ for Potenza, from -64.8 to -13.2 W m$^{-2}$ for Athens and from -90.3 to -28.4 W m$^{-2}$ for Limassol. The corresponding ranges of $\alpha_{532}$ are 0.286–0.029 km$^{-1}$, 0.268–0.088 km$^{-1}$, 0.135–0.078 km$^{-1}$ and 0.547–0.214 km$^{-1}$, respectively. Peaks in $\alpha_{532}$ are observed usually between 2 and 6 km a.s.l. indicating the intrusion of dust that corresponds to a decrease in the solar radiation reaching the surface.

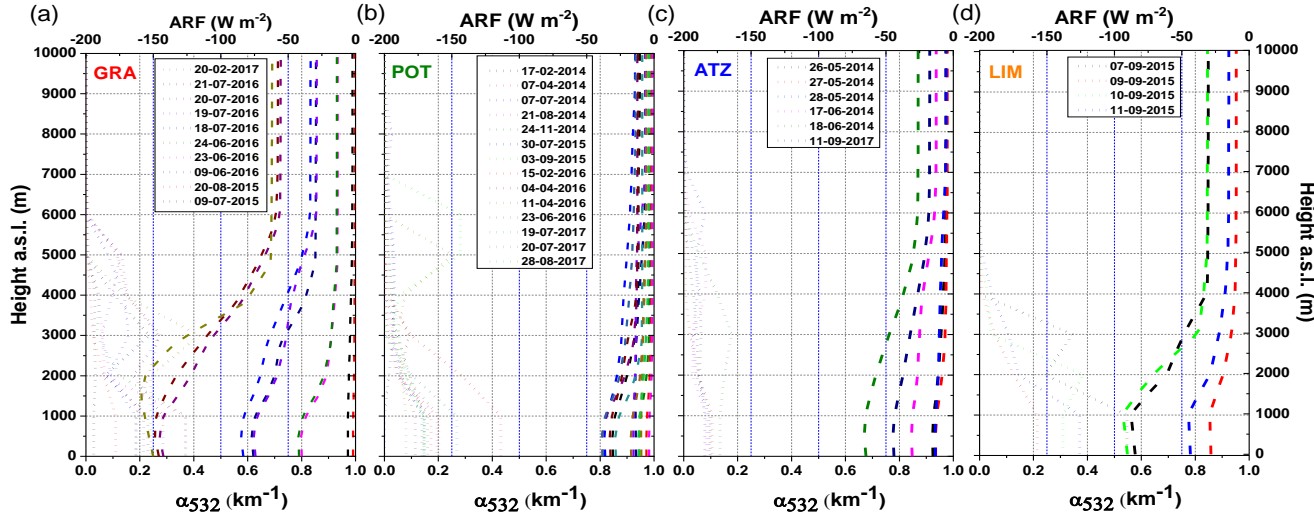

Figure 10: Vertical profiles of $\alpha_{532}$ (dotted lines) calculated from Raman lidar measurements along with the SW ARF (dashed lines) estimated from Libradtran simulations for the sites of: a) Granada, b) Potenza, c) Athens and d) Limassol, at 45° SZA.

## 5.    Conclusions

The characteristics of aerosol layers dominated by dust optical, geometrical, and radiative properties over the Mediterranean region were presented in this study. A total of 51 independent aerosol lidar measurements of Saharan dust events, studied over 4 southern European cities, were carefully selected and analyzed. The dust layers were usually observed between ~1.6 and ~5 km with $\delta_{p532}$ and $LR_{532}$ values ranging from 0.16 to 0.35 and from 35 to 75 sr respectively, depending on the air mass mixing state. Significantly high AOT$_{532}$ values ($0.40 \pm 0.31$) were found for Granada indicating that the dust outbreaks occurring over this area were more intense during the studied period. Results of $LR_{532}$ versus $\delta_{p532}$ are presented in order to elucidate the difference of pure dust and dust mixtures cases. Layers with lower $\delta_{p532}$ ($0.17 \pm 0.01$), AOT$_{532}$ ($0.03 \pm 0.02$) and thicknesses ($786 \pm 212$ m) values have shown high dust mixing ratio, while the properties of the least or no mixed dust layers ($\delta_{p532}$=0.32±0.02, AOT$_{532}$=0.32±0.23 and thickness=3158±1605 m) are in a good agreement with literature findings for pure Saharan dust cases (Tesche et al., 2009; Papayannis et al., 2009; Ansmann et al., 2012; Mona et al., 2012; Groß et al., 2011; 2013). Lidar stand-alone retrieved aerosol microphysical properties like the R$_{eff}$, RRI and IRI are also differentiated by the level of mixing.

Despite the numerous individual studies, the uncertainty in estimating the aerosols effect in climate change remains high. Therefore, coordinated and simultaneous studies using data from observation sites operating continuously, such as the EARLINET database are necessary for investigating the climatic effect of aerosols in a larger scale. Three Schemes have been implemented in our study to evaluate the ARF during the selected dust outbreaks: the model mass concentrations by BSC-DREAM8b (Scheme A), the vertical mass concentrations calculated from the dust-only component of the $\beta_{532}$ (Scheme B) and the $\alpha_{532}$ vertical profiles along with the mean AOT$_{532}$ values (Scheme C).

Lidar derived Schemes B and C are used here as input methods in LibRadtran simulations, since not many techniques have been widely used for retrieving the ARF using lidar vertical measurements as input. Their outputs are compared to the ones retrieved from Scheme A (based on BSC-DREAM8b model). On the one hand, Scheme B gives the opportunity to calculate only the DRF, even though many assumptions and constants are included in the calculation of the dust mass concentration values. On the other hand, Scheme C is more direct, since the $\alpha_{532}$ profiles are primarily used for retrieving the ARF in the SW range, but without providing a separation of dust and non-dust components. Consequently, the ARF values of Scheme C seem to be overestimated compared to those of Scheme B. These two implemented Schemes can contribute to the characterization of the aerosols' radiative forcing effects over the Mediterranean region, being one of the most sensitive regions to climate forcing (Kim et al., 2019). Scheme A is only recommended for cases were no lidar measurements are available but an estimation of the ARF is required, while one should take into account all the possible underestimations and a model such as BSC-DREAM8b includes.

The ARF variations are strong (of the order of 75%) and result from significant changes in the lidar retrieved optical properties due to the different intensities of the studied cases ($\alpha_{532}$, $\beta_{532}$, AOT$_{532}$) or the model mass estimations from the BSC-DREAM8b. Additional variations (of the order of 40%) in the SW range are introduced due to the variations in the available solar radiation during day (SZA). The vertical structure of a layer that provides information about the base, the thickness and the intensity (AOT) of a dust layer is critically important, while additional information of its mixing state can be also significant in ARF and net heating rate estimations. Our findings show a much more pronounced ARF at the BOA (ranging from -40 to -13 W m$^{-2}$ at SZA 25°, from -43 to -14 W m$^{-2}$ at SZA 45° and from -44 to -15 W m$^{-2}$ at 65°) compared to the one at the TOA (ranging from -9.5 to -1.4 W m$^{-2}$ at 25°, -16 to -3.3 W m$^{-2}$ at 45° and -24.3 to -6.9 W m$^{-2}$ at 65°) due to the low altitude of the studied layers (usually 2-4 km).

The systematic use of remote sensing vertical profiling measurements as input to radiative transfer models is stressed in this study, creating an essential tool allowing the estimation of the radiative effects produced by different aerosol types such as dust and its mixtures on a regional and a global scale. A further investigation of aerosols' mixing state is needed since, not only their optical but also their microphysical properties and radiative forcing can strongly vary, depending on the mixing types. Furthermore, we recommend the use of remote and in situ
measurements in the next generation state-of-the-art dust cycle models for the ARF should be intensified.

**Acknowledgements**

O. S.'s research has been financed through a scholarship from the General Secretariat for Research and Technology (GSRT) and the Hellenic Foundation for Research and Innovation (HFRI). A.P., R.F., and C.A.P. acknowledge support by the project "PANhellenic infrastructure for Atmospheric Composition and climatE
change" (MIS 5021516) which is implemented under the Action "Reinforcement of the Research and Innovation Infrastructure", funded by the Operational Programme "Competitiveness, Entrepreneurship and Innovation" (NSRF 2014-2020) and co-financed by Greece and the European Union (European Regional Development Fund). The EARLINET lidar data, were made available through the financial support by the ACTRIS Research Infrastructure Project funded by the European Union's Horizon 2020 research and innovation program under grant agreement no.
654169. The authors also acknowledge the BSC-DREAM8b model, operated by the Barcelona Supercomputing Center, the NOAA Air Resources Laboratory (ARL) for the provision of the HYSPLIT, the Google Earth and the AERONET for high-quality sun/sky photometer measurements. The Biomedical Research Foundation of the Academy of Athens (BRFAA) is acknowledged for the provision of its mobile platform to host the NTUA AIAS lidar system.

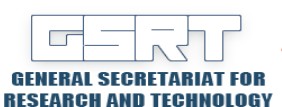 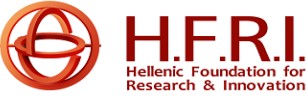 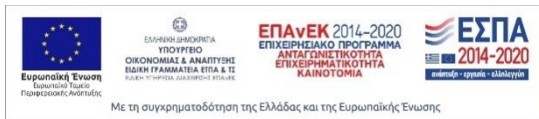

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
