# Peer review of "EARLINET observations of Saharan dust intrusions over the northern Mediterranean region (2014-2017): Properties and impact in radiative forcing"

_Atmospheric Chemistry and Physics, 2020_

## Short Comment (SC1) · 12 Jul 2020

A suggestion for making this study even better is to calculate the radiative effect within the atmospheric column as well. In this way, the results will be in the same reference with the IPCC radiative forcing that is calculated at the tropopause. A minor comment is to use the term "radiative effect" instead of "radiative forcing" since the latter refers to differences in irradiance compared to pre-industrial emissions. SInce in this study you compare dust aerosol scenarios to background aerosol scenarios, it would be more appropriate to use the term "radiative effect".

---

## Author Comment (AC1) · 24 Jul 2020

Thank you for your suggestions. It could be easy to include the RF within the atmospheric column as an extra plot or by replacing the RF at BOA and TOA respectively. There is a point in your comment about the term "radiative effect", however in the majority of the relative literature the term RF is widely used (e.g. Sicard et al., 2014; Meloni et al., 2003; 2015; Gkikas et al. 2018) and the results of this study are compared to these corresponding findings referred as RF. This is why the term RF was selected. Let us wait for the final comments of the reviewers.

---

## Referee Comment (RC1) · Anonymous Referee #1 · 25 Jul 2020

Authors analyze the dust outbreaks over Mediterranean basing on multiwavelengths lidar observations from four EARLINET sites. What is important, authors make next step: they use derived parameters of dust layers for estimation the dust direct forcing. Paper is well and clearly written and can be published in ACP. I have just technical comments.

*Ln.110 the $\alpha_{aer}$ and $\beta_{aer}$ vertical profiles, with systematic uncertainties of ~5–15% and ~10–25%, respectively*

Probably should be opposite. Uncertainty of backscattering is lower

Ln *112 mean uncertainty for $AE_\alpha$ and $AE_\beta$ is of order 7–21% and*

Providing uncertainty of Angstrom in percents makes no sense (What if it is zero?). Should be absolute values.

Ln 187 *The CRI grid was narrowed down to [1.4, 1.5]*

Why it was limited by 1.5? Real part can be higher

Ln 188 *and [0, 0.001, 0.005, 0.01] for the Imaginary part*

So only four values for Im were used?

Table 3 Authors should pay attention to uncertainty in this table

**RRI** | 1.50 = 0.00

What does it mean? No error? The same is for imaginary part. For some cases error of IRI is 10%. I doubt it. Imaginary part has spectral dependence. For what wavelength results are provided?

Lidar ratios at 355 nm (with corresponding uncertainties) are not provided in the table.

---

## Referee Comment (RC2) · Anonymous Referee #2 · 4 Aug 2020

The authors present a study using a combination of ground-based aerosol profiling and models. Statistics of lidar and depolarization ratio values, aerosol optical thicknesses, Ångström exponents, and geometrical properties of (mixed) Saharan dust layers over western, central, and eastern Mediterranean are reported. As the main result, the lidar measurements of the dust layers were used to calculate their radiative forcing with a radiative transfer model. The results were partly validated with ground-based radiation measurements. Additionally, using a conversion technique, the measured optical properties were used to retrieve microphysical properties (dust mass concentration

profiles), which were compared with a dust model.

The manuscript covers an important aspect of aerosol remote sensing with lidar, as the authors correctly state, namely using type separated aerosol profiles in e.g., dust forecast as well as radiative transfer models.

Nevertheless, I have some concerns which should be addressed before publishing (see comments below and in the attached pdf).

Major comments:

1. The authors state, that they estimate the dust radiative forcing based on 51 selected cases with respect to a clear sky background. But in fact, they calculate the aerosol radiative forcing of dusty aerosol layers- These aerosol layers are partly pure dust cases, but also dust cases mixed with other aerosols. Thus it cannot be claimed that the dust radiative forcing is calculated. As the authors use well-known techniques to separate the dust contribution in the observed lidar profiles, it would have been interesting and very innovative, to discuss if this dust-only profiles have could have also been used to really determine the dust radiative forcing. Then one could also discuss the contribution of the other aerosol types mixed with the Saharan dust, i.e. the non-dust radiative forcing.

2. Concerning the reported radiative forcing effects of the different schemes applied, there is too much simple reporting of values instead of a real discussion. So please address the following questions

2a. What conclusion do you draw from your three used schemes? The first conclusions is, that the Scheme A underestimated the mass concentration and thus radiative forcing. But what do we learn from the comparison of Scheme B and C?

2b. With respect to the evaluation with ground based radiation sensors: Is scheme B more correct than scheme C? Or the other way around, or no nothing of the both? Why did you use these three schemes?

[Figure]

2c. Are the layer geometrical properties from the model (Scheme A) equal to that of the lidar or have there been significant differences in the layer heights and extents also influencing the radiative forcing? Please discuss

2d. Concluding: You present the results from the 3 schemes intensively, but a proper discussion is missing. What is the most appropriate scheme, what are the weaknesses and strengths of the schemes and what is your recommendation for future research?

3. The language is partly sloppy, i.e. not clearly scientific, please see comments attached and check for the whole manuscript. I.e. put special emphasis when writing about dust, if you refer to dust-only properties or to aerosol layers containg dust.

4. Sometimes proper references are missing, see pdf.

5. Also the explanations are sometimes unclear and a rephrasing /extension is needed. See pdf as well. E.g. for the SphInX software tool

6. Minor, mainly textual suggestions, can be found in the attached pdf.

Considering the amount of minor comments, major revisions are needed. But I am confident that the authors can address all issues!

Specific comments:

1. Page 5, line 178:

"For LRd the mean values of $52 \pm 8$ sr, $51 \pm 9$, $52 \pm 9$ sr and $49 \pm 6$ sr were used per site, respectively as calculated from our findings."

So you used these lidar ratios as the pure dust lidar ratios in the conversion from optical to microphysical properties. Furthermore, you say you calculated them from your findings. I do not understand how. They are not a result of this conversion, they are an input parameter. Based on which criterion were they calculated? Are these the averages of only those layers having a large particle linear depolarization ratio of >0.31 at 532 nm wavelength? How many cases of such layers occurred and were analyzed?

Or are these the averages of all your cases? I suggest to describe this in more detail and make a reference to Sect. 4. There you present average values, but there you describe also mixed dust (dusty) cases, based on the criterion introduced on page 4, line 125.

2. Page 8, line 277:

"Before using the aerosol mass concentrations vertical profiles..."

Do you calculate and compare the whole aerosol mass concentration or only the mass concentration of the dust fraction? If it is the latter, you have to state that correctly. If it is the first, then you have to describe the calculation (list the conversion parameters) for the non-dust fraction, especially the used non-dust lidar ratio. In dusty layers with a particle linear depolarization ratio of 0.16 at 532 nm wavelength (the lower boundary of the selection criterion), a significant amount of non-dust mass can be expected.

3. A more detailed explanation of Fig. 3 is needed, in Caption and text.

Please also note the supplement to this comment:
https://acp.copernicus.org/preprints/acp-2020-611/acp-2020-611-RC2-supplement.pdf

**Supplement:**

[revised manuscript text omitted]

**Summary of Comments on acp-2020-611-s2_commented_2_test.pdf**

**Page: 1**
* * *
**Number: 1**     Author:     Date: 04.08.2020 11:15:06

In my opinion you need to rephrase the title. At the moment you state, that the Saharan dust intruded in the frame of EARLINET - which is of course not true. Thus, probably rephrase to something like: EARLINET observations of Saharan dust intrusions over the northern Mediterranean region (2014-2017): .....
* * *
**Number: 2**     Author:     Subject: ▮▮▮▮▮▮     Date: 27.07.2020 09:55:16

(space)
* * *
**Number: 3**     Author:     Subject: ▮▮▮▮▮▮     Date: 27.07.2020 09:55:16

(space)
* * *
**Number: 4**     Author:     Subject: ▮▮▮▮▮▮     Date: 27.07.2020 09:55:16

(space)
* * *
**Number: 5**     Author:     Subject: ▮▮▮▮▮▮     Date: 27.07.2020 09:55:16

I
* * *
**Number: 6**     Author:     Subject: ▮▮▮▮▮▮     Date: 27.07.2020 09:55:16

I
* * *
**Number: 7**     Author:     Subject: ▮▮▮▮▮▮     Date: 27.07.2020 09:55:16

R
* * *
**Number: 8**     Author:     Date: 04.08.2020 11:14:39
* * *
**Number: 9**     Author:     Subject: ▮▮▮▮▮▮     Date: 31.07.2020 15:21:19

agl or asl?
* * *
**Number: 10**     Author:     Subject: ▮▮▮▮▮▮     Date: 31.07.2020 15:23:29

I do not understand: Do you mean to identify which aerosols have mixed with dust? please rephrase.
* * *
**Number: 11**     Author:     Date: 04.08.2020 11:13:47

I suggest to use math minus $-$
* * *
**Number: 12**     Author:     Subject: ▮▮▮▮▮▮     Date: 26.07.2020 22:17:04

(locked space)
* * *
[Figure]

**1. Introduction**

The Saharan desert is one of the major dust sources globally, with dust advections to the Mediterranean countries to be modulated by meteorology along rather regular seasonal patterns (Mona et al., 2012). For instance, in the Western Mediterranean region, the African dust occurrence is higher in summer (Salvador et al., 2014), while in the central Mediterranean region, spring and summer are, usually, associated with dust aerosol loads extending up to altitudes of 3–4 km (Barnaba and Gobbi, 2004). In the Eastern Mediterranean, the main dust transport occurs from spring to autumn (Papayannis et al., 2009; Nisantzi et al., 2015; Soupiona et al., 2018) as a result of the high cyclonic activity over the northern Africa during these periods (Flaounas et al., 2015). Considering also that the Mediterranean basin is a region of high evaporation, low precipitation and remarkable solar activity, the transportation of aerosols accompanied with aging and mixing processes make this area a study of interest for present and future climate change effects (Michaelides et al., 2018).

It is well documented that mineral dust highly influences the atmospheric radiative balance through scattering and absorption processes (direct effect) as well as cloud nucleation, formation and lifetime (indirect effects) as shown in IPCC (2014). Considerable uncertainties in quantifying the global direct radiative effects of aerosols arise from the variability of their spatio-temporal distribution and the aging/mixing processes that can affect their optical, chemical and microphysical properties. Therefore, the magnitude and even the sign of the dust aerosol solar radiative forcing are highly uncertain as they strongly depend on their optical properties, their size distribution and their complex refractive index (CRI) values. Papadimas et al. (2012) reported that the aerosol optical depth seems to be the main parameter for modifying the regional aerosol radiative effects (under cloud-free conditions) and, that on an annual basis, aerosols can induce a significant "planetary" cooling over the broader Mediterranean basin. Other studies (Quijano et al., 2000; Tegen et al., 2010) have shown that the presence of clouds and the surface albedo are also unquestionable parameters affecting the net solar radiative transfer at the top of the atmosphere. However, a comprehensive analysis from ground-based aerosol optical properties to vertical profiles of short- and long-wave radiation estimations in the Mediterranean region has been reported so far only in a few papers (Sicard et al., 2014; Meloni et al., 2003; 2015; Gkikas et al. 2018).

Although there have been a lot of studies about Saharan dust optical properties based on the lidar technique (Papayannis et al., 2009; Mona et al., 2012; Navas-Guzmán et al., 2013; Soupiona et al., 2018), systematic long-term statistical studies are quite scarce since few aerosol depolarization data are available covering long periods. In Pérez et al. (2006) it is proposed that a regional atmospheric dust model, with integrated dust and atmospheric radiation modules, represents a promising approach for further improvements in numerical weather prediction practice and radiative impact assessment over dust-affected areas, especially in the Mediterranean. Hence, an in-depth study of the role of aerosols on the radiative forcing into different regions in the Mediterranean basin is still needed. While a synergy of ground-based lidar measurements and modelling seems very promising for obtaining radiative forcing estimations of dust aerosols, the use of inputs from regional models could also contribute for such estimations in areas where measurements are unavailable.

This paper aims to fill some of the aforementioned gaps by combining statistical lidar data of aerosol optical and microphysical properties with radiative transfer estimations and is organized as follows. A brief summary of the four selected EARLINET Mediterranean stations is given in Sect. 2 along with the data selection of the dust cases. Section 3 includes the description of the methodologies applied and the tools and models used for retrieving the aerosol optical and microphysical properties and their radiative forcing. The evaluation of the retrieved aerosol mass concentration profiles and of the ground level radiation are also presented. The results of the aerosol optical, geometrical and microphysical properties of the individual dust layers and the clusters, as well as the relevant radiative forcing calculations over the studied areas are discussed in Sect. 4. Finally, concluding remarks are given in Sect. 5.

**2. Instrumentation and data**

Number: 1     Author:     Subject: ███████     Date: 31.07.2020 15:23:35

'

Number: 2     Author:     Subject: ███████     Date: 27.07.2020 08:51:11

(this "--" is the right way to write from-to. use it everywhere instead of "-")

Number: 3     Author:     Subject: ███████     Date: 26.07.2020 22:17:04

summarized

Number: 4     Author:     Subject: █████████     Date: 31.07.2020 15:29:31

please provide also some references of measurements taken outside Europe. Both, near source and away from the source.

Number: 5     Author:     Subject: ████████     Date: 31.07.2020 15:29:45

Anything new concerning this topic since 2006?

Number: 6     Author:     Subject: ███████     Date: 26.07.2020 22:17:04

(space)

[revised manuscript text omitted]

Number: 1     Author:     Subject: ███████████     Date: 31.07.2020 15:31:02
any reference concerning EARLINET

Number: 2     Author:     Subject: ███████████     Date: 31.07.2020 15:31:15
Any reference?

Number: 3     Author:     Subject: ███████████     Date: 31.07.2020 15:32:22
reference or website?

Number: 4     Author:     Subject: ███████████     Date: 31.07.2020 15:34:43
This is not a complete sentence. Please rephrase, i.e. make 2 sentences out of it.

Number: 5     Author:     Subject: █████████     Date: 27.07.2020 09:55:31

Number: 6     Author:     Subject: ████████     Date: 31.07.2020 15:37:19
multiwavelength

Number: 7     Author:     Subject: ████████     Date: 31.07.2020 15:37:51
 at multiple wavelengths.

Number: 8     Author:     Subject: ████████     Date: 31.07.2020 15:37:57
N

Number: 9     Author:     Subject: █████████     Date: 26.07.2020 22:17:04
-

Number: 10     Author:     Subject: ███████████     Date: 31.07.2020 15:38:57
are you sure that you are meaning sytematic errors?

Number: 11     Author:     Subject: ████████     Date: 26.07.2020 22:17:04
?

Number: 12     Author:     Subject: ████████     Date: 26.07.2020 22:17:04
why not write west?

Number: 13     Author:     Subject: ███████████     Date: 31.07.2020 15:36:10
Checking the EARLIENT webpage, I see that the stations abbreviations listed there consist of 3 digits. Can you adapt this nomenclature? GRA, POT, ATZ, LIM

Number: 14     Author:     Subject: █████████     Date: 26.07.2020 22:17:04
m

Number: 15     Author:     Subject: ████████     Date: 26.07.2020 22:17:04
m

Number: 16     Author:     Subject: ████████     Date: 26.07.2020 22:17:04
m

Number: 17     Author:     Subject: ████████     Date: 26.07.2020 22:17:04
m

Number: 18     Author:     Subject: ████████     Date: 26.07.2020 22:17:04
discussed

Number: 19     Author:     Subject: ████████     Date: 26.07.2020 22:17:04
measured by lidar

Number: 20     Author:     Subject: ████████     Date: 31.07.2020 15:40:35
ion

Saharan dust events per station listed in the EARLINET database for the period 2014–2017, we considered for
125    further analysis only the data meeting three basic criteria: a) $\delta_{p532} \geq 0.16$ in the free troposphere, b) 35 sr $\leq$
LR $_{532} \leq 75$ sr in the free troposphere and c) the thickness of the detected layer to be 500 m, at least. The critical
height (in meters a.s.l.) in which the first criterion was met, was considered to be the base of the dust layer. This
assumption was deemed necessary to be made since usually, the lofted dust layers cannot be distinguished from the
top of the Planetary Boundary layer (PBL), while the presence of urban haze and pollution decreases drastically the
130    $\delta_p$ values down to 0.03–0.10 (Gobbi et al., 2000; Groß et al., 2013). The top of the dust layer was estimated as the
height where the signals were similar to the molecular scattering in the free troposphere. For some cases of the
Athens station, where depolarization measurements were unavailable, the values of the base and top were calculated
from the Raman lidar signals, following the procedure proposed by Mona et al. (2006).

Moreover, we performed a careful investigation of the air mass origin and dust transport path by means of
135    backward trajectory analysis. This analysis was carried out using the HYbrid Single-Particle Lagrangian Integrated
Trajectory (HYSPLIT) model (https://ready.arl.noaa.gov/HYSPLIT_traj.php, (Stein et al., 2015) together with the
GDAS (Global Data Analysis System) meteorological files (spatial resolution of 1° × 1°, every 3 hours) as data
input. The kinematic back-trajectories were calculated using the vertical velocity component given by the
meteorological model with a 96–120 hours pathway (4–5 days back). Modis/Terra information
140    (https://firms.modaps.eosdis.nasa.gov/map) was also taken into account for the corresponding hot spots of possible
fires and thermal anomalies along the trajectories (https://firms.modaps.eosdis.nasa.gov/map, not shown here).

Thus, we ended up with 51 individual cases in total (15 for Granada, 18 for Potenza, 12 for Athens and 6 for
Limassol). For the region of Cyprus, the situation is more complex since Middle East dust outbreaks occur also
frequently in addition to the Saharan dust events (Nisantzi et al., 2015; Kokkalis et al., 2018; Solomos et al., 2019).
145    On top of that, dust particles originating from Middle East proved to have different lidar ratio values than the
corresponding observations over Saharan desert (Mamouri et al., 2013; Kim et al., 2020). Taking this into account,
dust cases over the Limassol station originating from Middle East regions were excluded from our study.

The air mass trajectory analysis based on HYSPLIT for each station reveals the origin of each observed layer
(Fig. 1). In the majority of cases, air masses originate from the west and northwest Africa (Morocco, Mauritania,
150    Algeria and Tunisia). At a first glance, two occurrences seem to dominate: i) trajectories that travel directly from
the source to the observation stations and ii) trajectories that circulate over the Mediterranean or the Atlantic Ocean
(for the Granada cases), Europe and Balkans or even Turkey (for the Limassol cases) before reaching the
observation stations.

[Figure]

155

Figure 1: 96–120-hour backward trajectories for air masses arriving over a) Granada, b) Potenza, c) Athens and d) Limassol,
for arrival heights of approximately the center of each observed dust layer (51 cases in the period 2014–2017).

**Number: 1**    Author:    Subject: ███████████    Date: 31.07.2020 15:41:19

which signals? do you mean particle depolarization was close to 0 and particle backscatter as well?

**Number: 2**    Author:    Subject: ███████████    Date: 26.07.2020 22:17:04

**Number: 3**    Author:    Subject: ███████████    Date: 26.07.2020 22:17:04

(space)

**Number: 4**    Author:    Subject: ███████████    Date: 26.07.2020 22:17:04

,

**Number: 5**    Author:    Subject: █████████    Date: 31.07.2020 15:44:12

and Potenza

**Number: 6**    Author:    Subject: █████████    Date: 31.07.2020 15:43:55

and Athens

[Figure]

**3. Methodologies, tools and data evaluation**

In order to perform simulations for further investigating the behavior of the transported dust aerosols and their impacts we used different methods with a variety of tools and models. In this section we present our efforts for retrieving vertical dust mass concentration profiles, aerosol microphysical properties and radiative forcing results. Evaluation of the models used was also implemented.

**3.1. Dust mass concentration lidar retrievals**

To retrieve the aerosol dust mass concentration profiles, we used the $\beta_{532}$ and the $\delta_{p532}$ coefficients. Furthermore, by assuming that we have two aerosol types (dust and non-dust) inside the calculated $\beta_{532}$ values, we separated the backscatter profiles in two components: the first arising from the contribution of the weakly depolarizing particles ($\delta_{nd} = 0.05$ for non-dust particles) and the second from the contribution of strongly depolarizing particles ($\delta_d = 0.31$ for dust particles). Then, the dust-related backscatter coefficient $\beta_d$ at 532 nm was obtained, following the procedure described by Tesche et al. (2009). The estimation of the height-resolved mass concentration (in kg m$^{-3}$) of dust particles was based on the procedure described by Ansmann et al. (2012), using the following equation,

$$mass_d = \rho_d(v_d/\tau_d)\beta_d \, LR_d \qquad (1)$$

where in our study the coarse-particle mass density ($\rho_d$) was assumed equal to 2.6 gr cm$^3$ and a mean volume-to-AOT ratio for coarse mode particles, $v_d/\tau_d$ was calculated from AERONET measurements (https://aeronet.gsfc.nasa.gov/) for each station during the period 2014–2017. We report these values as follows: $0.71 \pm 0.37$ μm for Potenza, $0.80 \pm 0.29$ μm for Granada, $0.94 \pm 0.50$ μm for Athens, and $0.87 \pm 0.27$ μm for Limassol. The mean values of the whole studied period were calculated, since only few of the studied cases were common in EARLINET and AERONET database. For LR$_d$ the mean values of $52 \pm 8$ sr, $51 \pm 9$, $52 \pm 9$ sr and $49 \pm 6$ sr were used per site, respectively as calculated from our findings.

**3.2. The Spheroidal Inversion eXperiments (SphInX) software tool**

[revised manuscript text omitted]

**Number: 1**     Author:     Subject: ▌       Date: 26.07.2020 22:17:04
ied

**Number: 2**     Author:     Subject: ▌       Date: 26.07.2020 22:17:04
by

**Number: 3**     Author:     Subject: ▌       Date: 26.07.2020 22:17:04
see comments.

**Number: 4**     Author:     Subject: ▌       Date: 31.07.2020 16:27:13
and

**Number: 5**     Author:     Subject: ▌       Date: 31.07.2020 16:30:57
this I do not understand, which point do you mean?

**Number: 6**     Author:     Subject: ▌       Date: 31.07.2020 16:28:52
what is this? SD explained?

**Number: 7**     Author:     Subject: ▌       Date: 31.07.2020 16:29:14
what do you mean with simulation patterns?

**Number: 8**     Author:     Subject: ▌       Date: 04.08.2020 10:55:46
This is not evident for me, can you explain how this is seen?

**Number: 9**     Author:     Subject: ▌       Date: 31.07.2020 16:34:47
Is this seen in Figure 3?

**Number: 10**     Author:     Subject: ▌       Date: 31.07.2020 16:36:08
These considerations, however, would lead to an increased random error and no systematic effect like you have observed? Or am I wrong?

**Number: 11**     Author:     Subject: ▌       Date: 31.07.2020 16:36:43
Please extent caption.

**Number: 12**     Author:     Subject: ▌       Date: 31.07.2020 16:36:28
P

**Number: 13**     Author:     Subject: ▌       Date: 31.07.2020 16:37:06
.

were calculated in order to numerically quantify the performance of the global irradiance calculated from pyranometers and simulated from the three Schemes. Table 2 shows the statistical results for the modeled global irradiance values versus the reference pyranometer measurements for both locations and the threes schemes. All scheme simulations perform remarkably well, with rRMSE values ranging from 8.3 to 16.2% and rMBE values between 0 and 15.2%. In general, the rRMSE is slightly higher at Granada, mainly for the Scheme A. According to this statistic, the LibRadtran output with the best performance are those obtained with the Scheme C as input followed by Scheme B and A, respectively. This order is the same attending to the rMBE values with the exception of the Scheme A at Athens. The correlation coefficient r depicts the good performance of the radiative transfer model for the three schemes and the two locations models. All simulations present a $r > 0.95$ with minor differences (below a 10%) in the normalized SD values respect to the pyranometer global irradiance values. A slight overestimation is observed for all scheme outputs at Granada (norm SD > 1). Conversely, this overestimation is no longer evident in the modeled global irradiance for Athens. It is important to notice the good performance of the Scheme B despite the important calculus involved in it.

Table 2: Statistical metrics for the modeled global irradiance values versus the reference pyranometer measurements for Granada and Athens and the threes schemes applied.

| | Granada | | | | Athens | | | |
|---|---|---|---|---|---|---|---|---|
| | rRMSE (%) | rMBE (%) | r | SD (norm) | rRMSE (%) | rMBE (%) | r | SD (norm) |
| **Scheme A** | 16.2 | 15.2 | 0.99 | 1.09 | 10.8 | - 0.2 | 0.97 | 0.89 |
| **Scheme B** | 11.9 | 5.7 | 0.97 | 1.10 | 10.2 | 8.3 | 0.99 | 0.92 |
| **Scheme C** | 8.8 | 5.9 | 0.99 | 1.09 | 8.3 | 6.3 | 0.99 | 0.96 |

**4. Results**

**4.1. Aerosol geometrical and optical properties per site**

For each case studied, the mean $\delta_{p532}$, mean $LR_{532}$ and mean $AOT_{532}$ values were calculated inside the dust layers as shown in Fig. 4a-d). The corresponding standard deviation (SD) values give an indication of the variability of these parameters from base to the top of the dust layer. Figure 4a shows the aerosol geometrical properties for the detected layers one by one, per station and per year. The mean values of the base and top height of the dust layers per station, along with their SD are marked with horizontal bounded lines. At the four sites (Granada, Potenza, Athens and Limassol) mean layer thicknesses of $3392 \pm 1458$ m, $2150 \pm 1082$ m, $1872 \pm 816$ m and $1716 \pm 567$ m were calculated respectively. We also found mean $\delta_{p532}$ values of $0.24 \pm 0.05$, $0.26 \pm 0.06$, $0.28 \pm 0.05$ and $0.28 \pm 0.04$ (Fig. 4b), and mean $AOT_{532}$ values of $0.40 \pm 0.31$, $0.11 \pm 0.07$, $0.12 \pm 0.10$ and $0.32 \pm 0.17$ (Fig. 4d), respectively. Similar mean $LR_{532}$ values of around 51 sr (Fig. 4c) were found for all stations. The Granada station holds the minimum mean value for layers' base height ($1567 \pm 788$ m) and the maximum for top height ($4960 \pm 975$ m) and layer's thickness. Concerning the LR values no remarkable deviations were observed among the four stations having mean values around 50 sr, which are in very good agreement with literature findings (Tesche et al., 2009; Ansmann et al., 2012; Groß et al., 2011; 2013). The largest mean AOT value, equal to $0.40 \pm 0.31$, observed over Granada station is in accordance to the geometrical properties of Fig. 4 figure a that depicts thick dust layers in the majority of the cases.

Number: 1     Author:     Subject: ███████     Date: 26.07.2020 22:17:04
s

Number: 2     Author:     Subject: ███████     Date: 26.07.2020 22:17:04

Number: 3     Author:     Subject: ███████     Date: 27.07.2020 08:23:18
why is it important?

Number: 4     Author:     Subject: ███████     Date: 26.07.2020 22:17:04
(space)

Number: 5     Author:     Subject: ███████     Date: 31.07.2020 17:06:35
I guess asl, right?

Number: 6     Author:     Subject: ███████     Date: 26.07.2020 22:17:04

Number: 7     Author:     Subject: ███████     Date: 26.07.2020 22:17:04

Number: 8     Author:     Subject: ███████     Date: 26.07.2020 22:17:04
(

Number: 9     Author:     Subject: ███████     Date: 26.07.2020 22:17:04
)

(a)

(b)

(c)

(d)

[Figure]

Figure 4: Mean values along with standard deviation of a) base and top, b) $\delta_{p532}$, c) $LR_{532}$ and d) $AOT_{532}$, per station (text and banded lines) and per case (symbols and error bars) inside the observed dust layers.

Considering Granada's station as representative of the west Mediterranean region, Potenza of the central Mediterranean region, Athens and Limassol stations of the eastern Mediterranean region, a dust aerosol mode classification per region can be made. For this purpose, the mean $AOT_{532}$ versus the $AE_{\beta532/1064}$ giving an indication of the aerosol particle size in the atmospheric column, relevant to the aerosol load, for each region are shown in Fig. 5. A wide spread of the AOT values at moderate to low $AE_{\beta532/1064}$ values (between 0 and 0.6) observed in the Western Mediterranean region, demonstrates that the dust size distribution in this area is dominated by coarse mode particles during events of different intensities. On the other hand, the presence of dust layers, in the central and eastern Mediterranean regions can be associated with higher $AE_{\beta532/1064}$ values (even up to 1.5) and consequently, with the presence of fine mode particles and lower dust loads. Our findings verify that the longer the dust transport is, more likely it is for the dust aerosol properties to be modified, since dust can frequently be mixed with background aerosols in the eastern Mediterranean (Groß et al., 2019).

In terms of the aerosol size distribution, the scatter plot of Fig. 5 allowed us to perform a k-means clustering (Arthur and Vassilvitskii, 2007) in order to define three physically interpretable aerosol size distributions: a) fine mode, with $AE_{\beta532/1064} > 0.6$, b) coarse mode, with $AE_{\beta532/1064} \leq 0.6$ and $AOT_{532}$ between 0 and 0.2, c) whilst $AE_{\beta532/1064}$ values smaller than 0.6 attributed to large AOTs (between 0.2 and 0.8) are representative of extreme dust events. It seems that the majority of these extreme dust outbreaks occur over the Western Mediterranean region, more likely due to its location close to the African continent. For example, Fernández et al. (2019) recently reported an unprecedented extreme winter time Saharan dust event, during February 2017, over the whole Iberian Peninsula

Number: 1    Author:    Subject: ▮▮▮▮▮▮▮    Date: 31.07.2020 17:07:40
why is the particle size relevant for the aerosol load?

Number: 2    Author:    Subject: ▮▮▮▮▮▮▮    Date: 04.08.2020 10:59:33
are the dust particle properties really modified? I rather think it is  as you explain just later, that mixing is the major cause. Thus, I guess you mean the aerosol layer properties?

Number: 3    Author:    Subject: ▮▮▮▮▮▮▮    Date: 26.07.2020 22:17:04
the

[Figure]

360   with AOTs > 0.2 (675 nm) and *AE* values around zero. More studies referring to the occurrence of extreme dust events over the aforementioned area can be found in literature (Cachorro et al., 2008; Guerrero-Rascado et al., 2009).

[Figure]

Figure 5: $AE_{\beta 532/1064}$ versus $AOT_{532}$ per region: west (red marks and error bars), central (green marks and error bars) and east (blue marks and error bars) Mediterranean region. K-means clustering revealed three clusters: fine mode (ciel background), coarse mode (magenda background) and extreme dust events (yellow background).

**4.2. Clustering per mixing state**

Based on the High Spectral Resolution Lidar (HSRL) classification presented by Groß et al. (2013), we plotted the intensive aerosol quantities $LR_{532}$ versus $\delta_{p532}$ and we identified three of the six existing clusters in our data (Fig. 1. The first cluster (green marks and error bars) represents a mixing state of Saharan dust and BB aerosols having a large spreading in mean LR values and low mean $\delta_{p532}$ values (40 sr ≤ $LR_{532}$ ≤ 75 sr, 0.15 ≤ $\delta_{p532}$ ≤ 0.19). The second one, (red marks and error bars) is attributed to mixed Saharan dust, where dust aerosols are dominant, but urban/continental, marine or even pollen aerosols are also possibly present (40 sr ≤ $LR_{532}$ ≤65 sr, 0.20 ≤ $\delta_{p532}$ ≤ 0.29). The third cluster (orange marks and error bars) is attributed to pure Saharan dust aerosols (45 sr ≤ $LR_{532}$ ≤ 60 sr, 0.30 ≤ $\delta_{p532}$ ≤ 0.36). The most populated and consequently, the most common, among those three clusters is the red one, as expected, due to the frequent mixing of dust aerosols with continental ones (Papayannis et al, 2008). The range of our measured $\delta_{p532}$ values as indicated with the horizontal error bars in Fig. 2 overlap between the three identified aerosol clusters, showing a more realistic transition from one cluster to the other, bridging the gap specifically between green and red clusters from the HSRL classification.

| | Number: 1 | Author: | Subject: ▮▮▮▮▮▮ | Date: 26.07.2020 22:17:04 |
|---|---|---|---|---|

.

| | Number: 2 | Author: | Subject: ▮▮▮▮▮▮ | Date: 26.07.2020 22:17:04 |
|---|---|---|---|---|

(locked space)

[Figure]

Figure 6: $LR_{532}$ versus $\delta_{p532}$ values from HSRL observations presented by Groß et al. (2013), (colored dots) along with the selected datasets from the four EARLINET stations (symbols and error bars).

Table 3 summarizes the mean values of the aerosol geometrical, optical, and microphysical properties of the three identified clusters. A synergistic approach of HYSPLIT (trajectories of 120 hours backward for each case) and Google Earth (distance calculator) allowed us to estimate the distance travelled (in km) to the respective sites and the mixing hours per cluster. Specifically, the term of mixing refers to the hours the air masses travelled after leaving the African continent. We can see that the Saharan dust cluster of the air masses that present the lowest mixing with other air masses ($26 \pm 13$ hours), compared to the other clusters ($41 \pm 26$ hours for the BB & mixtures and $31 \pm 13$ hours for the mixed Saharan dust cluster). Moreover, the air masses of the Saharan dust cluster seem to travel faster than those of the other clusters, although covering a greater distance ($4845 \pm 2825$ km) at the same time (within 120 hours). Now, the main difference between the two remaining clusters (BB & mixtures/Mixed Saharan Dust) is attributed to the mixing hours. The air masses of the latter cluster remain 15 hours longer and circulate over the Mediterranean and Europe, so they are probably enriched with other types of aerosols.

Concerning the aerosol optical properties, the $\beta_{532}$ and $\alpha_{532}$ show lower values for BB & dust and for mixed Saharan dust cases ($1.10 \pm 0.15 \times 10^{-3}$ km$^{-1}$sr$^{-1}$, $0.47 \pm 0.28$ km$^{-1}$ and $1.24 \pm 0.80 \times 10^{-3}$ km$^{-1}$sr$^{-1}$, $0.74 \pm 0.48$ km$^{-1}$ respectively) and higher values ($1.54 \pm 1.05 \times 10^{-3}$ km$^{-1}$sr$^{-1}$, $0.80 \pm 0.27$ km$^{-1}$) for the Saharan dust cluster. Therefore, higher AOT values ($0.32 \pm 0.25$) were found for the latter cluster compared to the others, due to the higher dust burden of these events over the affected sites. The highest $\delta_{p532}$ values ($0.32 \pm 0.02$), indicate the arid origin and the coarse mode of pure Saharan dust layers (Freudenthaler et al., 2009) of the corresponding cluster. No direct information can be extracted from the similar $LR_{532}$ values about the mixing state of the aerosol layer, except that the range of the SD narrows as the mixing decreases. However, for the cases that observations at 355 nm were available, it seems that the lidar ratio color ratio (namely the $LR_{355}/ LR_{532}$) converges to unit for the Saharan Dust cluster, indicating the absence of spectral dependence for the case of pure dust (Müller et al., 2007; Veselovskii et al., 2020). For these cases also, the $AE_{\beta355/532}$ becomes closer to zero taking mean value of $0.35 \pm 0.45$.

We also summarise the changes in mean microphysical properties estimated with SphInX for the three identified clusters. The BB & Saharan dust cluster takes the lower mean $R_{eff}$ value ($0.293 \pm 0.074$ μm) due to the fine structure of BB aerosols included in the layer, while a mean $R_{eff}$ of $0.360 \pm 0.081$ μm corresponds to the cluster of mixed Saharan dust and a slightly larger value ($0.387 \pm 0.070$ μm) corresponds to the Saharan dust cluster. The values for RRI, IRI and SSA at 532 nm were similar for the two clusters that not include BB aerosols, whilst the presence of BB aerosols of the first cluster leads to higher RRI and IRI values and lower SSA, results

**Page: 12**

| Number: 1 | Author: | Subject: | Date: 26.07.2020 22:17:04 |
\text{x} or even \cdot

| Number: 2 | Author: | Subject: | Date: 26.07.2020 22:17:04 |
\text{x} or even \cdot

| Number: 3 | Author: | Subject: | Date: 26.07.2020 22:17:04 |
(locked space)

| Number: 4 | Author: | Subject: | Date: 26.07.2020 22:17:04 |
\text{x} or even \cdot

| Number: 5 | Author: | Subject: | Date: 31.07.2020 17:14:49 |
y

[revised manuscript text omitted]

Number: 1     Author:   Subject:        Date: 03.08.2020 13:12:09

Number: 2     Author:   Subject:        Date: 03.08.2020 13:12:06

Number: 3     Author:   Subject:        Date: 03.08.2020 13:11:43
,

Number: 4     Author:   Subject:        Date: 04.08.2020 11:00:10
did you calculate the dust DRF by using dust separated profiles only or the "total" profiles from your dusty cases? - see major comment

Number: 5     Author:   Subject:        Date: 26.07.2020 22:17:04
(locked space)

Number: 6     Author:   Subject:        Date: 26.07.2020 22:17:04

Number: 7     Author:   Subject:        Date: 03.08.2020 13:16:28
I do not understand this sentence.

Number: 8     Author:   Subject:        Date: 26.07.2020 22:17:04
(no space, no line break, use math minus)

Number: 9     Author:   Subject:        Date: 03.08.2020 13:17:11

Number: 10     Author:   Subject:        Date: 03.08.2020 13:17:02
I guess the dust aerosols are not at the top of the atmosphere....

Number: 11     Author:   Subject:        Date: 03.08.2020 13:18:02
at the TOA

reaching down to -34.8 W m$^{-2}$ at the surface in the Mediterranean area was also recently reported by Gkikas et al.
240    (2018) during the forecast period March 2000–February 2013.

Variations among these values are expected since they strongly depend on the different aerosol AOTs, mass estimations and extinction values. Estimations retrieved from Scheme B are expected to give higher values compared to those given from Scheme A as revealed also by Fig.3. The DRF at the LW spectral region is opposite in sign and significantly lower in absolute values than in the SW region. The difference between the TOA and BOA
445    DRF, with the former to be only weakly perturbed and the latter to be quite stronger, can be attributed to the heating within the troposphere, since the presence of the dust mainly leads to a displace of surface's radiative heating into the dust layer. Low reflected solar flux is partially offset by the absorption of upwelling LW radiation. Finally, in the LW spectral region, the mean DRF values at the BOA (Scheme A: +1.6±1.6 W m$^{-2}$, Scheme B: +4.6±4.7 W m$^{-2}$, and Scheme C: +2.9±9.4 W m$^{-2}$) are higher than those at the TOA (Scheme A: +0.8±0.9 W m$^{-2}$, Scheme B:
450    +3.6±4.4 W m$^{-2}$, and Scheme C: +1.2±6.2 W m$^{-2}$) due to the fact that the main source of LW radiation (Earth's surface) is close to the aerosol layers, mainly observed between 2 and 4 km a.s.l.. As a result, the RF$_{Atm}$ is positive during the diurnal circle, yielding net radiative heating of the dust layer.

[Figure]

The mean net heating rate within the atmosphere, calculated by adding algebraically both rates in the SW and LW spectral ranges is presented in Fig. 7b. Here, the net heating rate is clearly dependent on the available solar
455    radiation, that increases with SZA due to the low incoming solar radiation reaching the BOA at afternoon hours (SZA 65º). Our estimations are in accordance with the fact that as the SZA increases, the optical path of the SW radiation grows significantly, increasing the attenuation of the direct radiation while generating a higher fraction of the diffuse radiation. This effect is more pronounced at the BOA, in which, the intensity of the heating rate is reduced with increasing SZA, since fewer photons are available to heat the dust layers. The net heating rate values
460    for Scheme A are: -0.05±0.04 K day$^{-1}$ (25º), -0.04±0.03 K day$^{-1}$ (45º) and 0.00±0.02 K day$^{-1}$ (65º). Similar to slightly higher values are observed for Scheme B as follow: -0.07±0.06 K day$^{-1}$ (25º), -0.04±0.03 K day$^{-1}$ (45º), and -0.02±0.02 K day$^{-1}$ (65º). For Scheme C, we report higher values of this parameter during the diurnal circle. More precisely, the net heating rate is almost 1.5 times higher at 25º, 2 times higher at 45º and around 0.8 times higher at 65º, compared to the aforementioned Schemes. Greater sensitivity in the SZA appears in Scheme B as it results
465    from the line slope.

[Figure]

Figure 7: Mean values of a) SW and LW DRF at BOA and TOA and b) the net heating rate within the atmosphere, along with their SD for the three Schemes applied in the total set of the studied cases. The inserted box depicts the line slope.

In order to further explain the difference of sign in the net heating rate of Scheme C, compared to the two
470    others presented in Fig. 7b, we plotted the aforementioned parameter along with the base layer height, the AOT$_{532}$ and the layer thickness of each case as presented in Fig. 8. Taking into account that the effect of net heating rate of the vertical distributions, from negative to positive values, is more pronounced close to surface at small SZA values, the estimations of BOA at 25º SZA were selected to be presented in this graph. It becomes clear that the sign of the

**Number: 1**     Author:    Subject: ▮▮▮▮▮▮     Date: 03.08.2020 13:20:07
Please stay consistent and call it BOA.

**Number: 2**     Author:    Subject: ▮▮▮▮▮▮     Date: 04.08.2020 11:00:34
which forecast period? Never heard about that. A little more explanation about this paper is needed.

**Number: 3**     Author:    Subject: ▮▮▮▮     Date: 03.08.2020 13:20:40

**Number: 4**     Author:    Subject: ▮▮▮▮▮▮     Date: 04.08.2020 11:00:41
I do not understand this sentence.

**Number: 5**     Author:    Subject: ▮▮▮▮     Date: 03.08.2020 13:23:42
bad phrasing in this passage

**Number: 6**     Author:    Subject: ▮▮▮▮▮▮     Date: 03.08.2020 13:22:59
Is this statement significant if considering the uncertainties/variations?

**Number: 7**     Author:    Subject: ▮▮▮▮▮     Date: 26.07.2020 22:17:04
(locked space)

**Number: 8**     Author:    Subject: ▮▮▮▮▮▮     Date: 03.08.2020 13:24:07
please rephrase

**Number: 9**     Author:    Subject: ▮▮▮▮▮▮     Date: 03.08.2020 13:24:33
ever explained?

**Number: 10**     Author:    Subject: ▮▮▮▮▮▮     Date: 03.08.2020 14:07:46
please rephrase

**Number: 11**     Author:    Subject: ▮▮▮▮▮▮     Date: 03.08.2020 13:36:25
do you mean the estimation of the net heating rate at BOA?

net heating rate at BOA depends on the dust vertical structure and the AOT. More precisely, the majority of the
cases having low $AOT_{532}$ values ($\leq 0.2$) and low layer thickness ($\leq 2$ km) give negative net heating rate values.
Additionally, the higher the AOT the higher the absolute value of the net heating rate. Concerning the base layer
height, it plays a key role to the absolute net heating rate of each case, since dust layers close to the ground take
higher absolute net heating rate values. For example, a dust event with base of 2.8 km, 0.73 km thickness and
$AOT_{532}$ equal to 0.01 has a heating rate of -0.17 K day$^{-1}$, while a layer with base of 3.8 km, 0.66 km thickness and
$AOT_{532}$ equal to 0.02  has a net heating rate of almost zero (-0.03 K day$^{-1}$). Both the events occurred during the
same month (August). In another comparison, net heating rate values of -0.02 K day$^{-1}$ versus +0.09 K day$^{-1}$ are
reported for two layers, during summer time that have almost the same base (2.6 km and 2.5 km) and thickness (2.3
km and 2.4 km) but different $AOT_{532}$ values (0.08 and 0.34 respectively). Finally, a combination of high $AOT_{532}$
($0.21 - 0.83$) and high thickness ($2.1 - 5.5$ km), two parameters that are usually directly dependent, along with
low base height ($1.0 - 1.5$ km), give high net heating rate values with positive sign ranging from $+0.06$ to $+0.26$
K day$^{-1}$.

[Figure]

Figure 8: Net heating rate values per case of Scheme C estimated at BOA, 25° SZA versus base layer height. The horizontal
colorbar indicates the $AOT_{532}$ values and the vertical symbol thickness indicates the layer thickness.

Figures 9 (a-c) depicts the same results as in Fig. 7a but for each of the three identified clusters: BB and dust,
mixed Saharan dust Saharan dust. The DRF in the SW range is negative both in the BOA and TOA for all clusters
and is dominated by large dust particles of the Saharan dust episodes (Fig. 9c, Scheme A: -22.5±16.7 W m$^{-2}$, Scheme
B: -34.0±37.0 W m$^{-2}$, Scheme C: -49.2±50.9 m$^{-2}$ for BOA, and Scheme A: -2.5±2.1 W m$^{-2}$, Scheme B: -4.4±5.2 W
m$^{-2}$, Scheme C: -12.1±14.4 m$^{-2}$ for TOA, SZA 25°), whereas mixed layer with biomass burning aerosols have
significantly lower cooling effect (Fig. 9a, Scheme A: -6.2±4.0 W m$^{-2}$, Scheme B: -19±9 W m$^{-2}$, Scheme C: -4.8±3.5
W m$^{-2}$ for BOA, and Scheme A: -0.5±0.4 W m$^{-2}$, Scheme B: -2.0±1.3 W m$^{-2}$, Scheme C: -0.7±0.5 m$^{-2}$ for TOA, SZA
25°) for each of the three applied Schemes. DRF seems to be inversely proportional to the mixing ratio, since higher
absolute values are estimated for less mixed dust layers. This can directly be linked to the fact that RF values
strongly depend on $\alpha_{par}$, $\beta_{par}$ and AOT that take much higher values for the Saharan dust cluster as already reported
(Table 3). Focusing on the SW range, the cooling effect for Scheme A of the Saharan dust cluster is up to 3 times
higher compared to the BB and Saharan dust one, whilst the cooling effect for Scheme C of the former cluster is up
to 10 times higher compared to the latter. The cooling effect of Scheme B becomes also stronger with the decreasing
mixing state but in a lower magnitude (the former cluster is almost 2 times higher compared to the latter). Hence,
even though the cases included in the Saharan dust cluster usually take higher mass concentration values than the
other cases, as predicted by BSC-DREAM8b (Scheme A), still seemingly underestimates the intensity of strong

Number: 1        Author:    Subject:                        Date: 26.07.2020 22:17:04
(locked space)

Number: 2        Author:    Subject:                        Date: 26.07.2020 22:17:04
(locked space)

Number: 3        Author:    Subject:                  Date: 03.08.2020 14:13:49
 and

Number: 4        Author:    Subject:                        Date: 03.08.2020 14:15:33
This statement cannot be concluded from the Figure. How can it be concluded that the DRF is dominated by LARGE dust particles?

Number: 5        Author:    Subject:                        Date: 26.07.2020 22:17:04
(maybe locked space)

Number: 6        Author:    Subject:                        Date: 03.08.2020 14:17:34
please rephrase. Or do you mean dust layers mixed with biomass burning aerosols?

Number: 7        Author:    Subject:                        Date: 26.07.2020 22:17:04
(locked space)

Number: 8        Author:    Subject:                        Date: 26.07.2020 22:17:04
(locked space)

Number: 9        Author:    Subject:                        Date: 04.08.2020 11:01:33
This statement is only true if the "D" stands for direct and not for dust.

Number: 10       Author:    Subject:            Date: 03.08.2020 14:19:45

Number: 11       Author:    Subject:          Date: 03.08.2020 14:19:52

Number: 12       Author:    Subject:          Date: 03.08.2020 14:19:49

Number: 13       Author:    Subject:                  Date: 04.08.2020 11:01:50
to what corresponds "it"? please rephrase sentence.

transported dust episodes over the observation stations. On the contrary, Scheme C is the most sensitive to the mixing state. To explain this result one should consider that on the one hand,
spheroidal particles such as dust have larger surface area than spherical ones such as BB aerosols leading to larger AOTs (Haapanala et al., 2012) and consequently to increased negative DRF and on the other hand, the Schemes A and B involve greater assumptions concerning dust particles than Scheme C.

510

[Figure]

Figure 9: Mean values of SW and LW DRF at BOA and TOA and along with their SD for the three Schemes applied regarding the mixing state, namely a) BB & Saharan dust, b) Mixed Saharan dust and c) Saharan Dust. The dotted line represents the DRF zero value.

515      Finally, our interest is focused on the vertical DRF profiles from the surface (a.s.l.) up to 10 km height in the free troposphere, where airborne dust is usually found, as estimated by Scheme C at 45° SZA per station. The DRF profiles, in the SW region, presented in Figs. 10 (a-c) follow the aerosol extinction vertical structure. The DRF values at the BOA are high in absolute values with a cooling behavior and decreases with increasing height, while the magnitude is proportional to the aerosol load in the whole atmospheric column. Specifically, the DRF ranges

520      from -150.0 to -1.9 W m$^{-2}$ for Granada, from -38.1 to -3.7 W m$^{-2}$ for Potenza, from -64.8 to -13.2 W m$^{-2}$ for Athens and from -90.3 to -28.4 W m$^{-2}$ for Limassol. The corresponding ranges of $\alpha_{532}$ are 0.286–0.029 km$^{-1}$, 0.268–0.088 km$^{-1}$, 0.135–0.078 km$^{-1}$ and 0.547–0.214 km$^{-1}$, respectively. Peaks in $\alpha_{532}$ are observed usually between 2 and 6 km a.s.l. indicating the intrusion of dust that corresponds to a decrease in the solar radiation reaching the surface.

[Figure]

525      Figure 10: Vertical profiles of $\alpha_{532}$ (dotted lines) calculated from Raman lidar measurements along with the SW DRF (dashed lines) estimated from Libradtran simulations for the sites of: a) Granada, b) Potenza, c) Athens and d) Limassol, at 45° SZA.

Number: 1     Author:     Subject: ▮▮▮▮▮▮▮▮▮     Date: 04.08.2020 09:54:55
this is pure speculation. The size distribution of BB and dust aerosol is completely different.

Number: 2     Author:     Subject: ▮▮▮▮▮▮▮▮     Date: 26.07.2020 22:17:04
--

Number: 3     Author:     Subject: ▮▮▮▮▮▮▮▮     Date: 26.07.2020 22:17:04
(locked space)

[Figure]

**5.  Conclusions**

[revised manuscript text omitted]

**Page: 17**

Number: 1     Author:     Subject: ███████████     Date: 04.08.2020 11:04:27
aerosol layers dominated by dust

Number: 2     Author:     Subject: █████████     Date: 04.08.2020 11:04:44
 during our study period.

Number: 3     Author:     Subject: █████████     Date: 04.08.2020 11:05:12
What conclusion do you draw from your three used schemes? First conclusions is, that the Scheme A underestimated the mass concentration and thus radiative forcing. But what do  we learn from the comparison of Scheme B and C? -see major comment

Number: 4     Author:     Subject: ██████████     Date: 26.07.2020 22:17:04
(locked space)

Number: 5     Author:     Subject: ██████████     Date: 26.07.2020 22:17:04
(no line break)

Number: 6     Author:     Subject: █████████     Date: 26.07.2020 22:17:04

Number: 7     Author:     Subject: ██████████     Date: 04.08.2020 11:05:28
any reference for that?

Number: 8     Author:     Subject: ███████████     Date: 04.08.2020 11:05:45
I do not understand this sentence? you recomment modelling for dust models?

[Figure]

the European Union's Horizon 2020 research and innovation program under grant agreement no. 654169. The authors also acknowledge the BSC-DREAM8b model, operated by the Barcelona Supercomputing Center, the NOAA Air Resources Laboratory (ARL) for the provision of the HYSPLIT, the Google Earth and the AERONET for high-quality sun/sky photometer measurements. A.P., R.F., and C.A.P. acknowledge support by the project

575    "PANhellenic infrastructure for Atmospheric Composition and climatE change" (MIS 5021516) which is implemented under the Action "Reinforcement of the Research and Innovation Infrastructure", funded by the Operational Programme "Competitiveness, Entrepreneurship and Innovation" (NSRF 2014-2020) and co-financed by Greece and the European Union (European Regional Development Fund).

[Figure]

Number: 6  Author: Subject: ██████████  Date: 26.07.2020 22:17:04

Number: 7  Author: Subject: ██████████  Date: 26.07.2020 22:17:04
(space)

Number: 8  Author: Subject: ██████████  Date: 26.07.2020 22:17:04
Petzold, A.

Number: 9  Author: Subject: ██████████  Date: 26.07.2020 22:17:04
Atmos. Chem. Phys. 13(5)

Number: 10  Author: Subject: ██████████  Date: 26.07.2020 22:17:04
url?

Number: 11  Author: Subject: ██████████  Date: 26.07.2020 22:17:04
url?

Number: 12  Author: Subject: ██████████  Date: 26.07.2020 22:17:04

Number: 13  Author: Subject: ██████████  Date: 26.07.2020 22:17:04
--?

Number: 14  Author: Subject: ██████████  Date: 26.07.2020 22:17:04
is it online available? url?

[revised manuscript text omitted]

**Page: 20**

| Number: 1 | Author: | Subject: ███████████ | Date: 26.07.2020 22:17:04 |
|---|---|---|---|

is it online available? url?

| Number: 2 | Author: | Subject: ███████████ | Date: 26.07.2020 22:17:04 |
|---|---|---|---|

S

| Number: 3 | Author: | Subject: ███████████ | Date: 26.07.2020 22:17:04 |
|---|---|---|---|

| Number: 4 | Author: | Subject: ███████████ | Date: 26.07.2020 22:17:04 |
|---|---|---|---|

url?

| Number: 5 | Author: | Subject: ██████████ | Date: 26.07.2020 22:17:04 |
|---|---|---|---|

| Number: 6 | Author: | Subject: ███████████ | Date: 27.07.2020 08:30:26 |
|---|---|---|---|

. (check the correct abbreviation)

[revised manuscript text omitted]

Number: 1    Author:    Subject: ███████    Date: 26.07.2020 22:17:04

Number: 2    Author:    Subject: ███████    Date: 26.07.2020 22:17:04

Number: 3    Author:    Date: 04.08.2020 11:11:57

Number: 4    Author:    Subject: ███████    Date: 26.07.2020 22:17:04

Number: 5    Author:    Subject: ███████    Date: 26.07.2020 22:17:04

Number: 6    Author:    Subject: ███████    Date: 26.07.2020 22:17:04

Number: 7    Author:    Subject: ███████    Date: 27.07.2020 09:56:50
OAA

Number: 8    Author:    Subject: ███████    Date: 26.07.2020 22:17:04
but indeed, all dois should be displayed as urls anyway.

Number: 9    Author:    Subject: ███████    Date: 26.07.2020 22:17:04
url?

---

## Author Response (AR1)

**Answers to Reviewer 1**

The authors acknowledge the reviewer for the detailed and helpful comments that will allow us to improve this study. Our detailed respond to one-by-one reviewer's concerns is listed below (text in italics refer to the reviewer's comments while our response is with the blue text).

*Ln.110 the αaer and βaer vertical profiles, with systematic uncertainties of ~5–15% and ~10-25%, respectively. Probably should be opposite. Uncertainty of backscattering is lower*

Ln 110. (Line 119 in current version) Reviewer is right. This typo is now corrected.

Ln *112 mean uncertainty for AEα and* AEβ *is of order 7–21% and Providing uncertainty of Angstrom in percents makes no sense (What if it is zero?). Should be absolute values.*

Thank you. The Angstrom exponent uncertainty is given now in absolute values instead of %.

The lines 117-121 (current version) are revised as:

"By using the Raman technique, as proposed by Ansmann et al., (1992), we can retrieve the $\beta_{aer}$ and $\alpha_{aer}$ vertical profiles, with uncertainties of ~5–15% and ~10–25%, respectively (Ansmann et al., 1992; Mattis et al., 2002). Therefore, the corresponding uncertainty of the retrieved lidar ratio values is of the order of 11–30%, while the uncertainty for $AE_{\beta}$ and $AE_{\alpha}$ ranges between 0.02-0.04 and 0.03-0.08, respectively, as estimated by propagation error calculations."

*Ln 187 The CRI grid was narrowed down to [1.4, 1.5] Why it was limited by 1.5? Real part can be higher*

*Ln 188 and [0, 0.001, 0.005, 0.01] for the Imaginary part. So only four values for Im were used?*

Thank you for making this point. Below we respond to both concerns and we justify our CRI-grid selection.

When using Sphinx tool for spheroidal mode particles, the CRI grid in its most extended form can take on up to 42 pairs (7 RRIs x 6 IRIs) ([1.33, 1.4,1.5,1.6,1.7,1.8] x [0, 0.001, 0.005, 0.01, 0.03, 0.05, 0.1]).

For the cases presented in this study, we used only the 8 pairs for three main reasons:

1) Extreme absorption (e.g. for RRI=0.03 or 0.05) for dust particles is expected to manifest itself much less often. According to the literature, such values can be found either directly on dust site (see e.g. Schladitz et al., 2009) or e.g. when the dust concentration is lower so that a soot-type absorber prevails.

2) Preliminary runs with higher IRI and/or lower RRI have shown that the resulted shape-size distributions are less realistic, suggesting smoother representations and having undesired systematic behavior. This is indeed an inherent issue of the inversion process since high IRI values and/or low RRI values are known to smooth out the involved scattering cross sections, see e.g. (Samaras, S., 2016, Rother, 2009) and result in to more severely ill-posed problems (such as the one we are trying to solve here) raising the difficulty to solve for them. Thus, unless there are high enough levels of evidence in favor of including these values in the CRI grid, it seems logical to avoid them if possible.

3) Higher RRI values impose only a slight variation to the results, according to preliminary runs, and thus excluded to reduce the computational effort. The same methodology was also followed by Soupiona et al. (2019). The selected CRI grid [1.4, 1.5], [0, 0.001, 0.005, 0.01] is in good agreement with other literature findings (e.g. Benavent-Oltra et al., 2017).

Additionally, massive simulations performed by Samaras (2016) for different atmospheric scenarios showed that microphysical retrievals with an initially known CRI keep high accuracy and small uncertainty levels. Furthermore,

a variation of the RRI has a minor effect in the retrieved parameters $a_t$, $v_t$, $r_{eff}$ and a variation of IRI adds a relatively conservative percentage of 3-20% to the uncertainties compared to the fixed-CRI retrievals when the imposed measurement error is reasonably contained. For the retrieval of aerosol shape, the situation is more complicated, and simulations suggest that the quality of the results depend additionally on particle size. Regarding the influence of using different CRI grids, literature conclusively reports that it might have a severe impact even for the usual one-dimensional case (retrievals based on Mie theory). This is of course also the case for our approach here since it adds an additional dimension to the problem, i.e. the shape information, and simultaneously lacks finer and more extended radius- and aspect ratio ranges.

In the revised manuscript, the following text has been inserted (line 194-215), summarizing our aforementioned arguments as response to the reasonable concerns of both the reviewers:

"The SphInX software provides an automated process to carry out microphysical retrievals from synthetic and real lidar data inputs and further to evaluate statistically the inversion outcomes. It has been developed at the University of Potsdam (Samaras, 2016) within the Initial Training for atmospheric Remote Sensing (ITaRS) project (2012–2016). SphInX operates with expendable pre-calculated discretization databases based on spline collocation and on look-up tables of scattering efficiencies using T-matrix theory (Rother and Kahnert, 2009). This is to avoid the computational cost which would otherwise limit the microphysical retrieval to an impractical point. The methodology applied here for spheroid-particle approximation is the same as presented in Soupiona et al. (2019). Raman lidar observations were used as inputs for specific heights within the layers and averaged to produce the 6-point dataset of the so-called $3\beta_{par} + 2\alpha_{par} + 1\delta$ setup. All cases fulfilling this setup were treated in parallel for retrieving their microphysical properties. The Complex Refractive Index (CRI) is fed to the software separately for the real and imaginary parts which then constitutes a grid combining the following default values: Real part (RRI) $[1.33, 1.4, 1.5, 1.6, 1.7, 1.8]$ and Imaginary part (IRI) $[0, 0.001, 0.005, 0.01, 0.03, 0.05, 0.1]$. A range of values for the effective radius ($R_{eff}$), which occurs from the ratio of the total volume concentration ($u_t$) and the total surface-area concentration($a_t$), $r_{eff} = 3\,{}^{u_t}/_{a_t}$, is also needed to be predefined. Here, the effective radius $R_{eff}$ ranged between 0.01 μm and 2.2 μm and the CRI grid was narrowed down to $[1.4, 1.5]$ for the Real part (RRI), and $[0, 0.001, 0.005, 0.01]$ for the Imaginary part (IRI), in order to avoid retrieving less realistic size distributions that suggest smoother representations and have undesired systematic behavior. Indeed, Samaras (2016) showed for different atmospheric scenarios that microphysical retrievals with an initially known CRI keep high accuracy and small uncertainty levels. Furthermore, variations of the RRI have minor effects in the retrieved parameters and variations of the IRI adds a relatively conservative percentage of 3-20% to the uncertainties compared to the fixed-RI retrievals when the imposed measurement error is reasonably contained. The outputs presented here are the RRI and IRI, the Single Scattering Albedo (SSA) and the $R_{eff}$. "

*Table 3: Authors should pay attention to uncertainty in this table What does it mean? No error? The same is for imaginary part. For some cases error of IRI is 10%. I doubt it. Imaginary part has spectral dependence. For what wavelength results are provided? Lidar ratios at 355 nm (with corresponding uncertainties) are not provided in the table.*

The authors agree with the reviewer that the values presented in Table 3 (changed to Table 4 in current version) are misleading and clarification should be added in the text.

Firstly, the % quantities concerning the microphysical properties (including RRI and IRI) that are calculated by Sphinx, refer to retrieval uncertainties, and not to validations of the retrieved parameters from direct measurements. In Sphinx, the Variability (Var %) of a parameter stands for the standard deviation of the selected best (least-residual) values divided by their mean value. The Var (%) derived by using the 5 best solutions of the ill-posed problem as retrieved by the inversion algorithm. This is why the terms variability/uncertainty (Var/Unc %) are preferred when defining the terms. The uncertainties are expected to be relatively low since the aerosol layers were carefully chosen for the inversion by keeping AE/LR variation low and furthermore the preliminary runs did much

of the heavy lifting ruling out too incompatible solutions. However, one should keep in mind that Var refers to each case separately and not to the clusters and hence, it is not shown in this Table.

Secondly, Table 3 (changed to Table 4 in current version) shows the averaged values of $r_{eff}$, RRI, IRI and SSA along with their standard deviation for each cluster (5 cases for BB & dust, 8 cases for Sah. dust, 29 cases for mixed Sah. dust). Therefore, for instance $1.50 \pm 0.00$ "error" in the RRI means that among the 5 cases of the BB & dust cluster there is no variability in this parameter. Consequently, zero standard deviation is calculated.

Lidar ratios at 355 nm (with corresponding standard deviations) are now added in the updated Table.

In order to make this point clear to the reader and to avoid any misleading, in the revised manuscript the following text has been inserted (Lines 429-431):

"Table 4 summarizes the mean values of the aerosol geometrical, optical, and microphysical properties of the three identified clusters along with their SD (5 cases for BB & Saharan dust, 8 cases for Saharan dust, 29 cases for mixed Saharan dust)."

Also, at lines 453-454: "We also summarise the changes in mean microphysical properties estimated with SphInX tool for all the cases of each of the three identified clusters."

And the caption of Table 4 is updated as follows:

"Table 4: Mean values of optical, geometrical and microphysical properties of the three identified clusters along with their standard deviation (SD). Zero SD indicates no variability in the corresponding retrieved parameter. The term of mixing refers to the hours the air masses travelled after leaving the African continent."

**References**

Benavent-Oltra, J.A., Román, R., Granados-Munõz, M.J., Pérez-Ramírez, D., OrtizAmezcua, P., Denjean, C., Lopatin, A., Lyamani, H., Torres, B., Guerrero-Rascado, J.L., Fuertes, D., Dubovik, O., Chaikovsky, A., Olmo, F.J., Mallet, M., AladosArboledas, L., 2017. Comparative assessment of GRASP algorithm for a dust event over Granada (Spain) during ChArMEx-ADRIMED 2013 campaign. Atmos. Meas. Tech. 10, 4439–4457. https://doi.org/10.5194/amt-10-4439-2017.

Rother, T. Electromagnetic Wave Scattering on Nonspherical Particles. Springer, New York, 2009.

Samaras, S. Microphysical retrieval of non-spherical aerosol particles using regularized inversion of multi-wavelength lidar data. PhD Thesis, Institut für Mathematik, Numerische Mathematik - Inverse Probleme, University of Potsdam, Germany, 2016

Schladitz, A., Müller, T., Kaaden, N., Massling, A., Kandler, K., Ebert, M., Weinbruch, S., Deutscher, C., and Wiedensohler. A. In situ measurements of optical properties at Tinfou (Morocco) during the Saharan Mineral Dust Experiment SAMUM 2006. Tellus B, 61(1):64–78, 2009

Soupiona, O., Samaras, S., Ortiz-Amezcua, P., Böckmann, C., Papayannis, A., Moreira, G. A., Benavent-Oltra, J. A., Guerrero-Rascado, J. L., Bedoya-Velásquez, A. E., Olmo, F. J., Román, R., Kokkalis, P., Mylonaki, M., Alados-Arboledas, L., Papanikolaou, C. A. and Foskinis, R.: Retrieval of optical and microphysical properties of transported Saharan dust over Athens and Granada based on multi-wavelength Raman lidar measurements: Study of the mixing processes, Atmos. Environ., 214(July), 116824, doi:10.1016/j.atmosenv.2019.116824, 2019

**Answers to Reviewer 2**

The authors present a study using a combination of ground-based aerosol profiling and models. Statistics of lidar and depolarization ratio values, aerosol optical thicknesses, Ångström exponents, and geometrical properties of (mixed) Saharan dust layers over western, central, and eastern Mediterranean are reported. As the main result, the lidar measurements of the dust layers were used to calculate their radiative forcing with a radiative transfer model. The results were partly validated with ground-based radiation measurements. Additionally, using a conversion technique, the measured optical properties were used to retrieve microphysical properties (dust mass concentration profiles), which were compared with a dust model. The manuscript covers an important aspect of aerosol remote sensing with lidar, as the authors correctly state, namely using type separated aerosol profiles in e.g., dust forecast as well as radiative transfer models. Nevertheless, I have some concerns which should be addressed before publishing (see comments below and in the attached pdf file).

The authors are thankful to the reviewer for the detailed and helpful comments aiming to improve this study. Our detailed respond to one-by-one reviewer's concerns is listed below (text in black refer to the reviewer's comments while our response is with the blue text. The changes in the manuscript are mentioned with "…" and the corresponding line numbers are mentioned).

Major comments:

1. The authors state, that they estimate the dust radiative forcing based on 51 selected cases with respect to a clear sky background. But in fact, they calculate the aerosol radiative forcing of dusty aerosol layers- These aerosol layers are partly pure dust cases, but also dust cases mixed with other aerosols. Thus, it cannot be claimed that the dust radiative forcing is calculated. As the authors use well-known techniques to separate the dust contribution in the observed lidar profiles, it would have been interesting and very innovative, to discuss if this dust-only profiles have could have also been used to really determine the dust radiative forcing. Then one could also discuss the contribution of the other aerosol types mixed with the Saharan dust, i.e. the non-dust radiative forcing.

Thank you for addressing this point. The Dust Radiative Forcing (DRF) has now been changed to Aerosol Radiative Forcing (ARF). Concerning the dust contribution to the radiative forcing calculations (actually, in Scheme B), only the dust vertical distribution is used as input, (based on the separation of the $\beta_{532}$ into dust and non-dust contribution that led to the calculation of the vertical distribution of the dust-only mass concentration) in order to determine the dust radiative forcing. Scheme A also refers to the dust mass concentration as estimated by BSC-DREAM8b over the studied sites. On the other hand, in Scheme C both contributions of dust and non-dust aerosols (total $\alpha_{aer}$) are taken into account. This is the main difference of Scheme C compared to other two aforementioned Schemes (A and B) and this was one of the main reasons of presenting and analyzing all three Schemes, since each of them gives a different perspective to the same parameter studied here, the radiative forcing. Moreover, since the chemical composition of the non-dust components is unknown, an estimation of the non-dust mass concentration is hard to be done and exceeds the purpose of the present study. However, Figure 9 aims to give an estimation of how the other aerosol types mixed with the Saharan dust contribute in the radiative effect. A future study though, could be done examining the differences in radiative forcing among different aerosol types such as dust, biomass and anthropogenic aerosols. Multiple cases of aerosol layers having particle loads of different origins could then be examined. The AERONET inversion products could also provide further information about the asymmetry parameter, the single scattering albedo, the complex refractive index and other useful parameters that help for a deep investigation of the radiative forcing. However, we do not have currently common dates between EARLINET lidar and AERONET data, with the relevant information however provided by the latter to refer to the total atmospheric column.

Consequently, new lines that further clarify the usage of the three different Schemes have now been added in lines 273-285:

"The flowchart in Fig. 2 depicts these three Schemes applied to create the input files for the dust-loaded atmospheric conditions used in LibRadtran software package (Emde et al., 2016). Scheme A refers to the dust mass concentration as estimated by BSC-DREAM8b over the studied sites. In Scheme B, only the dust vertical distribution is used as input, (based on the separation of the $\beta_{532}$ into dust and non-dust components that led to the calculation of the vertical distribution of the dust-only mass concentration, Eq. 1) in order to determine the Dust Radiative Forcing (DRF). On the other hand, in Scheme C both contributions of dust and non-dust aerosols (total $\alpha_{aer}$) are taken into account. Additionally, for Scheme C conversion factors from OPAC were used in order to convert the $\alpha$par and the corresponding AOT from 532 nm to 10 μm (peak, within the atmospheric window). The conversion was based on an adaptive inversion algorithm of Shang et al. (2018) who presented a way to convert extinction coefficients at different wavelengths by using Ångström exponent values derived from AOTs. It should be mentioned here that the Scheme B, even though it also includes many assumptions and uncertainties in its calculations, is the only one, compared to the rest two (Schemes A and C) that gives us the opportunity to calculate only the dust contribution in the radiative effect."

2.  Concerning the reported radiative forcing effects of the different schemes applied, there is too much simple reporting of values instead of a real discussion. So please address the following questions

A concluding answer to Comments 2a, 2b and 2d, is given after comment 2d (see below) due to relevance. Taking into account the changes that have been made in the manuscript answering to the Reviewers' comments 1-2(a,b,c,d), the authors are now confident that the usage of the three different Schemes is justified and also the weaknesses and strengths of each one of them are clearer.

2a. What conclusion do you draw from your three used schemes? The first conclusion is that the Scheme A underestimated the mass concentration and thus radiative forcing. But what do we learn from the comparison of Scheme B and C?

2b. With respect to the evaluation with ground-based radiation sensors: Is scheme B more correct than scheme C? Or the other way around, or no nothing of the both? Why did you use these three schemes?

The main reason for performing this comparison was to check the performance of the model and that our simulations were close to reality. In Lines 354-364 (current version) it is mentioned that:

"All Scheme simulations perform remarkably well, with rRMSE values ranging from 8.3 to 16.2% and rMBE values between 0 and 15.2%. In general, the rRMSE is slightly higher at Granada, mainly for Scheme A. According to this statistic, the LibRadtran outputs with the best performance are those obtained by Scheme C as input followed by Schemes B and A, respectively. This order is the same attending to the rMBE values with the exception of Scheme A at Athens. The correlation coefficient r depicts the good performance of the radiative transfer model for the three schemes and the two locations. All simulations present a value of $r > 0.95$ with minor differences (below a 10%) in the normalized SD values respect to the pyranometer global irradiance values. A slight overestimation is observed for all scheme outputs at Granada (norm SD > 1). Conversely, this overestimation is no longer evident in the modeled global irradiance for Athens. However, it is important to note the good performance of the Scheme B despite the high number of various parameters involved in it."

However, no further conclusions can be extracted at this point, since many aspects, such as the location, the distance between the lidar and the pyranometers, the limited number of cases included in this specific comparison (data only for Athens and Granada stations) and other possible rapidly changing atmospheric parameters introduce uncertainties and may lead to hasty conclusions.

2c. Are the layer geometrical properties from the model (Scheme A) equal to that of the lidar or have there been significant differences in the layer heights and extents also influencing the radiative forcing? Please discuss

A brief comment was firstly mentioned comparing the layer geometrical properties between the model (Scheme A) and the lidar (lines 290-292 and lines 296-298 in the previous version, however, after the Reviewer's point of view, a separate paragraph was added (lines 338-348 in current version) as follow:

"By further comparing the modeled mass vertical profiles to the ones calculated by lidar, we report that the mean center of mass (in km) estimated from the BSC-DREAM8b profiles is about 0.6 km lower than the one calculated from the lidar measurements ($2.6 \pm 1.0$ km and $3.2 \pm 1.1$ km respectively). The maximum concentration (peak) is usually found in the height region 2-3 km, regarding both the predicted and the observed data. The BSC-DREAM8b, having a significantly lower vertical resolution compared to the lidar, predicts smoother profiles of dust layers by spreading the layer's base to lower altitudes (~1km, in 100% of the cases) and the top at higher altitudes (in 86% of the cases) compared to the observed ones. These remarks are in line with the previous studies of Mona et al. (2014) and Binietoglou et al. (2015) where they have reported discrepancies concerning the base, the top layer height and extinction profiles and good agreement in terms of profile shape, between the BSC-DREAM8b and observations. However, due to degradation of the spatial resolution in order to fit the fixed height levels of the OPAC dataset for the ARF simulations, these discrepancies in height were smoothed out."

2d. Concluding: You present the results from the 3 schemes intensively, but a proper discussion is missing. What is the most appropriate scheme, what are the weaknesses and strengths of the schemes and what is your recommendation for future research?

The abovementioned paragraph (see answer to comment 1) that has been added in Section 3.4.1. within the lines 273-285 is now appropriate to justify our choice to utilize the three different Schemes.

Moreover, the "Conclusions" section has now been changed in order to strengthen and enrich the reason of the use of the 3 different Schemes applied and the concluding remarks occurred from their comparison (Lines 593-611):

"Despite the numerous individual studies, the uncertainty in estimating the aerosols effect in climate change remains high. Therefore, coordinated and simultaneous studies using data from observation sites operating continuously, such as the EARLINET database are necessary for investigating the climatic effect of aerosols in a larger scale. Three Schemes have been implemented in our study to evaluate the ARF during the selected dust outbreaks: the model mass concentrations by BSC-DREAM8b (Scheme A), the vertical mass concentrations calculated from the dust-only component of the $\beta_{532}$ (Scheme B) and the $\alpha_{532}$ vertical profiles along with the mean $AOT_{532}$ values (Scheme C).

Lidar derived Schemes B and C are used here as input methods in LibRadtran simulations, since not many techniques have been widely used for retrieving the ARF using lidar measurements as input. Their outputs are compared to the ones retrieved from Scheme A (based on BSC-DREAM8b model). On the one hand, Scheme B gives the opportunity to calculate only the DRF, even though many assumptions and constants are included in the calculation of the dust mass concentration values. On the other hand, Scheme C is more direct, since the $\alpha_{532}$ profiles are primarily used for retrieving the ARF in the SW range, but without providing a separation of dust and non-dust components. Consequently, the ARF values of Scheme C seem to be overestimated compared to those of Scheme B. These two implemented Schemes can contribute to the characterization of the aerosols' radiative forcing effects over the Mediterranean region, being one of the most sensitive regions to climate forcing (Kim et al., 2019). Scheme A is only recommended for cases were no lidar measurements are available but an estimation of the ARF is needed, while one should take into account all the possible underestimations and a model such as BSC-DREAM8b includes."

3. The language is partly sloppy, i.e. not clearly scientific, please see comments attached and check for the whole manuscript. I.e. put special emphasis when writing about dust, if you refer to dust-only properties or to aerosol layers containg dust.
4. Sometimes proper references are missing, see pdf.
5. Also, the explanations are sometimes unclear and a rephrasing /extension is needed. See pdf as well. E.g. for the SphInX software tool

Answer to comments 3.4.5: Thank you for addressing these points. All suggested corrections and improvements in the attached PDF file have been applied in the text. The paragraph that includes the description of SphInX is now extended covering the concerns of both Reviewers (Lines 194-215):

"The SphInX software provides an automated process to carry out microphysical retrievals from synthetic and real lidar data inputs and further to evaluate statistically the inversion outcomes. It has been developed at the University of Potsdam (Samaras, 2016) within the Initial Training for atmospheric Remote Sensing (ITaRS) project (2012–2016). SphInX operates with expendable pre-calculated discretization databases based on spline collocation and on look-up tables of scattering efficiencies using the T-matrix theory (Rother and Kahnert, 2009). This is to avoid the computational cost which would otherwise limit the microphysical retrieval to an impractical point. The methodology applied here for spheroid-particle approximation is the same as presented in Soupiona et al. (2019). Raman lidar observations were used as inputs for specific heights within the layers and averaged to produce the 6-point dataset of the so-called $3\beta_{par} + 2\alpha_{par} + 1\delta$ setup. All cases fulfilling this setup were treated in parallel for retrieving their microphysical properties. The Complex Refractive Index (CRI) is fed to the software separately for the real and imaginary parts which then constitutes a grid combining the following default values: Real part (RRI) $[1.33, 1.4, 1.5, 1.6, 1.7, 1.8]$ and Imaginary part (IRI) $[0, 0.001, 0.005, 0.01, 0.03, 0.05, 0.1]$. A range of values for the effective radius ($R_{eff}$), which occurs from the ratio of the total volume concentration ($u_t$) and the total surface-area concentration($a_t$), $r_{eff} = 3\,{u_t}/{a_t}$, is also needed to be predefined. Here, the effective radius $R_{eff}$ ranged between 0.01 μm and 2.2 μm and the CRI grid was narrowed down to $[1.4, 1.5]$ for the Real part (RRI), and $[0, 0.001, 0.005, 0.01]$ for the Imaginary part (IRI), in order to avoid retrieving less realistic size distributions that suggest smoother representations and have undesired systematic behavior**.** Indeed, Samaras (2016) showed for different atmospheric scenarios that microphysical retrievals with an initially known CRI keep high accuracy and small uncertainty levels. Furthermore, variations of the RRI have minor effects in the retrieved parameters and variations of the IRI adds a relatively conservative percentage of 3-20% to the uncertainties compared to the fixed-RI retrievals when the imposed measurement error is reasonably contained. The outputs presented here are the RRI and IRI, the Single Scattering Albedo (SSA) and the $R_{eff}$."

6. Minor, mainly textual suggestions, can be found in the attached pdf.

All suggested improvements and corrections have been applied in the text.

Specific comments:

1. Page 5, line 178:

"For LRd the mean values of 52 ± 8 sr, 51 ± 9, 52 ± 9 sr and 49 ± 6 sr were used per site, respectively as calculated from our findings."

$$mass_d = \rho_d (v_d/\tau_d)\beta_d\, LR_d$$

So you used these lidar ratios as the pure dust lidar ratios in the conversion from optical to microphysical properties. Furthermore, you say you calculated them from your findings. I do not understand how. They are not a result of this conversion, they are an input parameter. Based on which criterion were they calculated? Are these the averages of only those layers having a large particle linear depolarization ratio

of >0.31 at 532 nm wavelength? How many cases of such layers occurred and were analyzed? Or are these the averages of all your cases? I suggest to describe this in more detail and make a reference to Sect. 4. There you present average values, but there you describe also mixed dust (dusty) cases, based on the criterion introduced on page 4, line 125.

The text has now been updated in order to be clearer (Lines 177-192, current version). A Table including the values that were previously mentioned in the text, is now inserted as it was suggested by the Reviewer.

"The estimation of the height-resolved mass concentration (in kg m$^{-3}$) of dust particles was based on the procedure described by Ansmann et al. (2012), using the following equation

$$mass_d = \rho_d (v_d / \tau_d) \beta_d\, LR_d \quad (1)$$

where in our study the coarse-particle mass density ($_{\rho d}$) was assumed equal to 2.6 g m$^{-3}$ and a mean volume-to-AOT ratio for coarse mode particles, $v_d / \tau_d$ was calculated from AERONET measurements (https://aeronet.gsfc.nasa.gov/) for each station during the period 2014-2017. Table 2 summarizes these values for the entire studied period, since only few cases were common in EARLINET and AERONET database. Regarding the LR$_d$ parameter, the mean LR values per station, as calculated from the lidar measurements, were used (cf. Table 2; Fig. 4c). These values are in good agreement with literature findings for long-range transported Saharan dust events (Tesche et al., 2009; Ansmann et al., 2012; Groß et al., 2011; 2013).

Table 2: Assumed ($\rho_d$) and computed parameters ($v_d / \tau_d$, LR$_d$) used for the estimation of the height-resolved mass concentration (in kg m$^{-3}$) of dust particles. The ratio $v_d / \tau_d$ is derived from AERONET sun–sky photometer measurements within the period 2014-2017 at Granada, Potenza, Athens and Limassol. The LR$_d$ is calculated from the available corresponding lidar measurements per station."

| Station | $\rho_d$ (g m$^{-3}$) | $v_d / \tau_d$ (μm) (AERONET) | LR$_d$ (sr) |
|---------|-----------------------|-------------------------------|-------------|
| GRA | 2.6 | 0.80±0.29 | 52±8 |
| POT | 2.6 | 0.71±0.37 | 51±9 |
| ATZ | 2.6 | 0.94±0.50 | 52±9 |
| LIM | 2.6 | 0.87±0.27 | 49±6 |

2. Page 8, line 277:

"Before using the aerosol mass concentrations vertical profiles: : :" Do you calculate and compare the whole aerosol mass concentration or only the mass concentration of the dust fraction? If it is the latter, you have to state that correctly. If it is the first, then you have to describe the calculation (list the conversion parameters) for the non-dust fraction, especially the used non-dust lidar ratio. In dusty layers with a particle linear depolarization ratio of 0.16 at 532 nm wavelength (the lower boundary of the selection criterion), a significant amount of non-dust mass can be expected.

The Reviewer is right, this sentence was misleading. For Scheme B, only the mass concentration of the dust fraction was taken into account. The updated sentence has now changed:

Lines 315-316: "Before using the vertical dust mass concentrations profiles retrieved from i) BSC-DREAM8b model simulations (Scheme A) and ii) lidar measurements as calculated from Eq. 1 (mass$_d$), (Scheme B) as inputs to the LibRadtran model, …"

Also, some relevant changes have been applied in Section 3.4.1, Lines 262-272 in order to be more specific about the three Schemes and our calculations:

"A set of four simulations was carried out per case of the studied dust events. The first two simulations refer to clear-sky atmosphere with background/baseline aerosol conditions (default properties: rural type aerosol in the boundary layer, background aerosol above 2 km, spring-summer conditions and a visibility of 50 km, index "clear" in Eq. 2), the first for the SW and the second for LW range, since these ranges are treated separately by LibRadtran. The remaining two simulations correspond to dust loaded atmosphere, again, the one for the SW range and the other for the LW range, respectively, for which the vertical profiles of the dusty layers were used as additional inputs (index "dusty" in Eq. 2). These inputs have been obtained with three different Schemes: A) vertical mass concentration profiles simulated by the BCS-DREAM8b model, B) vertical dust mass concentration profiles of only the dust component as calculated from Eq. 1 (mass$_d$) utilizing the $\beta_{532}$ coefficient and, C) vertical profiles of $\alpha_{532}$ along with the respective mean AOT$_{532}$ value. In the final step, we calculated the parameters $\Delta F$, ARF, ARF$_{NET}$ and ARF$_{ATM}$ applying Eqs. 2-5."

3. A more detailed explanation of Fig. 3 is needed, in Caption and text.

The caption has now been extended as follows:

"Taylor's diagram of the case-by-case vertical mass concentration simulated by BSC-DREAM8b model against the lidar retrieved ones. The black point (1,0) represent the calculated lidar data. The azimuthal angle presents the correlation coefficient (r), the radial distance of any point from the origin (0,0) indicates the normalized SD of the data set. The colored the dots represent each one of the 4 EARLINET stations, namely GRA (red), POT (green), ATZ (blue) and LIM (orange)."

Line 287, This is not evident for me, can you explain how this is seen?

The sentence has now been changed (lines 325-328 in updated version):

"In the 66 % of the cases there is a good correlation (r > 0.6), and consequently a good prediction of the shape of the vertical distribution is achieved, while in 96 % of the cases the model gives lower concentration values (normalized SD < 1) revealing an underestimation in the intensity and the mass concentration of the events."

Line 345, "relevant to the aerosol load ..." why is the particle size relevant for the aerosol load?

This expression was deleted since it was a bit misleading.

**Highlighted manuscript (changes are highlighted in yellow)**

[revised manuscript text omitted]
$^{-1}$) | $1.10 \pm 0.15$ [x10$^{-3}$] | $1.24 \pm 0.80$ [x10$^{-3}$] | $1.54 \pm 1.05$ [x10$^{-3}$] |
| | $\alpha_{532}$ (km$^{-1}$) | $0.47 \pm 0.28$ | $0.74 \pm 0.48$ | $0.80 \pm 0.27$ |
| | LR$_{532}$ (sr) | $51 \pm 15$ | $50 \pm 7$ | $52 \pm 5$ |
| | LR$_{355}$ (sr) | $35 \pm 13$ | $42 \pm 7$ | $51 \pm 10$ |
| | $\delta_{p532}$ | $0.17 \pm 0.01$ | $0.26 \pm 0.03$ | $0.32 \pm 0.02$ |
| | LR$_{355}$/LR$_{532}$ | $0.69 \pm 0.24$ | $0.84 \pm 0.16$ | $0.98 \pm 0.16$ |
| | AE$_{\beta355/532}$ | $0.44 \pm 0.59$ | $0.52 \pm 0.61$ | $0.35 \pm 0.45$ |
| | AOT$_{532}$ | $0.03 \pm 0.02$ | $0.15 \pm 0.10$ | $0.32 \pm 0.25$ |
| Geometry & Mixing | Thickness (km) | $0.79 \pm 0.21$ | $2.08 \pm 0.76$ | $3.10 \pm 1.72$ |
| | Distance (km) | $3496 \pm 1185$ | $3662 \pm 1617$ | $4845 \pm 2825$ |
| | Mixing (hours) | $41 \pm 26$ | $66 \pm 41$ | $26 \pm 13$ |
| Microphysical Properties | R$_{eff}$ ($\mu$m) | $0.293 \pm 0.074$ | $0.360 \pm 0.081$ | $0.387 \pm 0.070$ |
| | RRI | $1.50 \pm 0.00$ | $1.47 \pm 0.05$ | $1.47 \pm 0.05$ |
| | IRI | $0.005 \pm 0.000$ | $0.0046 \pm 0.0045$ | $0.0041 \pm 0.0018$ |
| | SSA$_{532}$ | $0.948 \pm 0.002$ | $0.964 \pm 0.018$ | $0.964 \pm 0.022$ |
| | SSA$_{355}$ | $0.937 \pm 0.007$ | $0.958 \pm 0.022$ | $0.952 \pm 0.026$ |

[revised manuscript text omitted]

---

## Author Response (AR2)

The authors have done a great job considering my comments. The manuscript has significantly improved. However, two minor clarification are needed before it can be published:

The authors are thankful to the reviewer for the helpful comments. Our detailed respond to reviewer's minor concerns is listed below (text in black refer to the reviewer's comments while our response is with the blue text).

1. Line 347: "However, due to degradation of the spatial resolution in order to fit the fixed height levels of the OPAC dataset for the ARF simulations, these discrepancies in height were smoothed out." I do not understand this sentence. So you did not use the layer heights derived from model and lidar, but fixed ones from the OPAC data set? Please clarify!

The reviewer is correct. The rephrased sentence (Lines 348-350) is now clearly written as below:

"However, since fixed height levels of the OPAC dataset were finally used in LibRadtran for the ARF simulations of the three Schemes, having significantly lower vertical resolution compared to the intitial lidar profiles, these discrepancies in height were smoothed out".

2. I think the description of SphInX still need improvement:

a. "..to carry out microphysical retrievals from synthetic and real lidar data inputs.."

I guess what you mean is to carry out calculations from lidar data to obtain microphysical aerosol properties? Is this correct, the please write it more precise!

The reviewer is right. There were syntactical errors in this sentence. Now the sentence is updated:

"The SphInX software provides an automated process to carry out calculations from lidar data for obtaining the aerosol microphysical properties and further to statistically evaluate the inversion outcomes".

b. "CRI grid was narrowed down to [1.4, 1.5] for the Real part (RRI), and [0, 0.001, 0.005, 0.01]". What do the values in the brackets mean? Do you only use 2 values for the RRI and 4 for the IRI? Please explain correctly without using mathematic notations which have not been explained.

c. Can you justify, why one can narrow down the CRI to the grid you use? Is this based on the properties of the aerosol type?

d. "...in order to avoid retrieving less realistic size distributions that suggest smoother representations and have undesired systematic behavior. "

What does this sentence mean? And what is an undesired behaviour? It sounds like data manipulation....please write more precisely.

Answer to the comments 2.b,c,d:

The corresponding lines have now been changed trying to fulfill the Reviewers' questions.

Here, the  $R_{eff}$  ranged between 0.01 µm and 2.2 µm and the CRI grid was narrowed down to include only the values 1.4 and 1.5 for the Real part (RRI), and the values 0, 0.001, 0.005, and 0.01 for the Imaginary part (IRI), providing a total of 8 possible combinations for the CRI grid (instead of the initial total of 42 CRI grid). These ranges were used after a careful investigation of the values of the aerosol optical and microphysical properties found in the literature concerning transported Saharan dust events (Dubovik et al., 2006; Weinzierl et al., 2011; Mishra et al., 2014; Benavent-Oltra et al., 2017; Veselovskii et al., 2016; Veselovskii et al., 2020) in order to avoid retrieving less realistic dust-related size distributions and CRI values and to minimize the computational time.

e. "imposed measurement error is reasonably contained"

What does this mean? Please state correctly.

All in all, I still have not really understood what and how you have done the calculations of the aerosol microphysical properties. Please revise this paragraph.

Actually, this sentence was aiming to compare the selection of a narrow CRI grid to a totally fixed CRI value, as described in Samaras (2016), but we now believe that a reference to the "fixed CRI" case is out of the scope of the description that was implemented here, and hence, the total sentence was deleted.

The total updated paragraph is presented below (Lines 194-216):

The SphInX software provides an automated process to carry out calculations from lidar data for obtaining the aerosol microphysical properties and further to statistically evaluate the inversion outcomes. It has been developed at the University of Potsdam (Samaras, 2016) within the Initial Training for atmospheric Remote Sensing (ITaRS) project (2012–2016). SphInX operates with expendable pre-calculated discretization databases based on spline collocation and on look-up tables of scattering efficiencies using T-matrix theory (Rother and Kahnert, 2009). This is to avoid the computational cost which would otherwise limit the microphysical retrieval to an impractical point. The Complex Refractive Index (CRI) is fed to the software separately for the real and imaginary parts which then constitutes a grid combining the following default values: Real part (RRI) [1.33, 1.4, 1.5, 1.6, 1.7, 1.8] and Imaginary part (IRI) [0, 0.001, 0.005, 0.01, 0.03, 0.05, 0.1]. A range of values for the effective radius (Reff), which occurs from the ratio of the total volume concentration (ut) and the total surfacearea concentration(at),  $r_{eff} = 3^{u_t}/a_{t}$ , is also needed to be predefined. The methodology applied here for spheroid-particle approximation is the same as presented in Soupiona et al. (2019). More specifically, hourly Raman lidar measurements were used as inputs for specific heights within the observed dusty layers and were averaged to produce the 6-point dataset of the so-called  $3\beta_{par} + 2\alpha_{par} + 1\delta$  setup. All cases fulfilling this setup were treated in parallel for retrieving their microphysical properties. Here, the Reff ranged between 0.01 µm and 2.2 µm and the CRI grid was narrowed down to include only the values 1.4 and 1.5 for the Real part (RRI), and the values 0, 0.001, 0.005, and 0.01 for the Imaginary part (IRI), providing a total of 8 possible combinations for the CRI grid (instead of the initial total of 42 CRI grid). These ranges were used after a careful investigation of the values of the aerosol optical and microphysical properties found in the literature concerning transported Saharan dust events (Dubovik et al., 2006; Weinzierl et al., 2011; Mishra et al., 2014; Benavent-Oltra et al., 2017; Veselovskii et al., 2016; Veselovskii et al., 2020) in order to avoid retrieving less realistic dust-related size distributions and CRI values and to minimize the computational time. The outputs presented here are the RRI and IRI, the Single Scattering Albedo (SSA) and the Reff.